# IN SEARCH OF LOST DOMAIN GENERALIZATION

**Ishaan Gulrajani**[*]
Stanford University
igul222@gmail.com

**David Lopez-Paz**
Facebook AI Research
dlp@fb.com

## ABSTRACT

The goal of domain generalization algorithms is to predict well on distributions different from those seen during training. While a myriad of domain generalization algorithms exist, inconsistencies in experimental conditions—datasets, network architectures, and model selection criteria—render fair comparisons difficult. The goal of this paper is to understand how useful domain generalization algorithms are in realistic settings. As a first step, we realize that model selection is non-trivial for domain generalization tasks, and we argue that algorithms without a model selection criterion remain incomplete. Next we implement DOMAINBED, a testbed for domain generalization including seven benchmarks, fourteen algorithms, and three model selection criteria. When conducting extensive experiments using DO-MAINBED we find that when carefully implemented and tuned, ERM outperforms the state-of-the-art in terms of average performance. Furthermore, no algorithm included in DOMAINBED outperforms ERM by more than one point when evaluated under the same experimental conditions. We hope that the release of DOMAINBED, alongside contributions from fellow researchers, will streamline reproducible and rigorous advances in domain generalization.

## 1 INTRODUCTION

Machine learning systems often fail to *generalize out-of-distribution*, crashing in spectacular ways when tested outside the domain of training examples (Torralba and Efros, 2011). The overreliance of learning systems on the training distribution manifests widely. For instance, self-driving car systems struggle to perform under conditions different to those of training, including variations in light (Dai and Van Gool, 2018), weather (Volk et al., 2019), and object poses (Alcorn et al., 2019). As another example, systems trained on medical data collected in one hospital do not generalize to other health centers (Castro et al., 2019; AlBadawy et al., 2018; Perone et al., 2019; Heaven, 2020). Arjovsky et al. (2019) suggest that failing to generalize out-of-distribution is failing to capture the causal factors of variation in data, clinging instead to easier-to-fit spurious correlations prone to change across domains. Examples of spurious correlations commonly absorbed by learning machines include racial biases (Stock and Cisse, 2018), texture statistics (Geirhos et al., 2018), and object backgrounds (Beery et al., 2018). Alas, the capricious behaviour of machine learning systems out-of-distribution is a roadblock to their deployment in critical applications.

Aware of this problem, the research community has spent significant efforts during the last decade to develop algorithms able to generalize out-of-distribution. In particular, the literature in *Domain Generalization (DG)* assumes access to multiple datasets during training, each of them containing examples about the same task, but collected under a different *domain* or experimental condition (Blanchard et al., 2011; Muandet et al., 2013). The goal of DG algorithms is to incorporate the invariances across these training domains into a classifier, in hopes that such invariances will also hold in novel test domains. Different DG solutions assume different types of invariances, and propose algorithms to estimate them from data.

Despite the enormous importance of DG, the literature is scattered: a plethora of different algorithms appear yearly, each of them evaluated under different datasets, neural network architectures, and model selection criteria. Borrowing from the success of standardized computer vision benchmarks

---

[*] Alphabetical order, equal contribution. Work done while IG was at Facebook AI Research. This paper is a living benchmark, always refer to the latest version available at https://arxiv.org/abs/2007.01434

Table 1: Our ERM baseline outperforms the state-of-the-art in terms of average domain generalization performance, even when picking the best competitor per dataset.

| Dataset / algorithm | DG accuracy per test domain | | | | | | Average |
|---|---|---|---|---|---|---|---|
| Rotated MNIST (full) | 0° | 15° | 30° | 45° | 60° | 75° | |
| DIVA (Ilse et al., 2019) | 95.3 | 98.7 | 98.7 | 98.4 | 97.7 | 94.5 | 97.2 |
| Our ERM | 95.9 | 98.9 | 98.8 | 98.9 | 98.9 | 96.4 | **98.0** |
| VLCS | C | L | S | V | | | |
| G2DM (Albuquerque et al., 2019) | 95.5 | 67.6 | 69.4 | 71.1 | | | 75.9 |
| Our ERM | 97.7 | 64.3 | 73.4 | 74.6 | | | **77.5** |
| PACS | A | C | P | S | | | |
| RSC (Huang et al., 2020) | 87.9 | 82.1 | 97.9 | 83.4 | | | **87.8** |
| Our ERM | 84.7 | 80.8 | 97.2 | 79.3 | | | 85.5 |
| OfficeHome | A | C | P | R | | | |
| DDAIG (Zhou et al., 2020) | 59.2 | 52.3 | 74.6 | 76.0 | | | 65.5 |
| Our ERM | 61.3 | 52.4 | 75.8 | 76.6 | | | **66.5** |
| All datasets | | | | | | | |
| Best SOTA competitor | | | | | | | 81.6 |
| Our ERM | | | | | | | **81.9** |

such as ImageNet (Russakovsky et al., 2015), the purpose of this work is to perform a rigorous comparison of DG algorithms, as well as to open-source our software for anyone to replicate and extend our analyses. This manuscript investigates the question: *How useful are different DG algorithms when evaluated in a consistent and realistic setting?*

To answer this question, we implement and tune fourteen DG algorithms carefully, to compare them across seven benchmark datasets and three model selection criteria. There are three major takeaways from our investigations:

- **Claim 1:** A careful implementation of ERM outperforms the state-of-the-art in terms of average performance across common benchmarks (Table 1, full list in Appendix A.5).
- **Claim 2:** When implementing fourteen DG algorithms in a consistent and realistic setting, no competitor outperforms ERM by more than one point (Table 3).
- **Claim 3:** Model selection is non-trivial for DG, yet affects results (Table 3). As such, we argue that DG algorithms should specify their own model selection criteria.

As a result of our research, we release DOMAINBED, a framework to streamline rigorous and reproducible experimentation in DG. Using DOMAINBED, adding a new algorithm or dataset is a matter of a few lines of code. A single command runs all experiments, performs all model selections, and auto-generates all the result tables included in this work. DOMAINBED is a living project: we welcome pull requests from fellow researchers to update the available algorithms, datasets, model selection criteria, and result tables.

Section 2 kicks off our exposition with a review of the DG setup. Section 3 discusses the difficulties of model selection in DG and makes recommendations for a path forward. Section 4 introduces DOMAINBED, describing the features included in the initial release. Section 5 discusses the experimental results of running the entire DOMAINBED suite, illustrating the competitive performance of ERM and the importance of model selection criteria. Finally, Section 6 offers our view on future research directions in DG. Appendix A reviews one hundred articles spanning a decade of research in DG, summarizing the experimental performance of over thirty algorithms.

Table 2: Learning setups. $L^d$ and $U^d$ denote the labeled and unlabeled distributions from domain $d$.

| Setup | Training inputs | Test inputs |
|---|---|---|
| Generative learning | $U^1$ | $\emptyset$ |
| Unsupervised learning | $U^1$ | $U^1$ |
| Supervised learning | $L^1$ | $U^1$ |
| Semi-supervised learning | $L^1, U^1$ | $U^1$ |
| Multitask learning | $L^1, \ldots, L^{d_{\mathrm{tr}}}$ | $U^1, \ldots, U^{d_{\mathrm{tr}}}$ |
| Continual (or lifelong) learning | $L^1, \ldots, L^\infty$ | $U^1, \ldots, U^\infty$ |
| Domain adaptation | $L^1, \ldots, L^{d_{\mathrm{tr}}}, U^{d_{\mathrm{tr}}+1}$ | $U^{d_{\mathrm{tr}}+1}$ |
| Transfer learning | $U^1, \ldots, U^{d_{\mathrm{tr}}}, L^{d_{\mathrm{tr}}+1}$ | $U^{d_{\mathrm{tr}}+1}$ |
| **Domain generalization** | $L^1, \ldots, L^{d_{\mathrm{tr}}}$ | $U^{d_{\mathrm{tr}}+1}$ |

## 2 THE PROBLEM OF DOMAIN GENERALIZATION

The goal of supervised learning is to predict values $y \in \mathcal{Y}$ of a target random variable $Y$, given values $x \in \mathcal{X}$ of an input random variable $X$. Predictions $\hat{y} = f(x)$ about $x$ originate from a predictor $f : \mathcal{X} \to \mathcal{Y}$. We often decompose predictors as $f = w \circ \phi$, where we call $\phi : \mathcal{X} \to \mathcal{H}$ the featurizer, and $w : \mathcal{H} \to \mathcal{Y}$ the classifier. To solve the prediction task we collect *the* training dataset $D = \{(x_i, y_i)\}_{i=1}^n$, which contains identically and independently distributed (i.i.d.) examples from the joint probability distribution $P(X, Y)$. Given a loss function $\ell : \mathcal{Y} \times \mathcal{Y} \to [0, \infty)$ measuring prediction error, supervised learning seeks the predictor minimizing the risk $\mathbb{E}_{(x,y) \sim P}[\ell(f(x), y)]$. Since we only have access to the data distribution $P(X, Y)$ via the dataset $D$, we instead search a predictor minimizing the *empirical* risk $\frac{1}{n} \sum_{i=1}^n \ell(f(x_i), y_i)$ (Vapnik, 1998).

The rest of this paper studies the problem of *Domain Generalization (DG)*, an extension of supervised learning where training datasets from multiple domains (or environments) are available to train our predictor (Blanchard et al., 2011). Each domain $d$ produces a dataset $D^d = \{(x_i^d, y_i^d)\}_{d=1}^{n_d}$ containing i.i.d. examples from some probability distribution $P(X^d, Y^d)$, for all training domains $d \in \{1, \ldots, d_{\mathrm{tr}}\}$. The goal of DG is *out-of-distribution generalization*: learning a predictor able to perform well at some unseen test domain $d_{\mathrm{tr}} + 1$. Since no data about the test domain is available during training, we must assume the existence of statistical invariances across training and testing domains, and incorporate such invariances (but nothing else) into our predictor. The type of invariance assumed, as well as how to estimate it from the training datasets, varies between DG algorithms. We review a hundred articles in DG spanning a decade of research and thirty algorithms in Appendix A.5.

DG differs from unsupervised domain adaptation. In the latter, unlabeled data from the test domain is available during training (Pan and Yang, 2009; Patel et al., 2015; Wilson and Cook, 2018). Table 2 compares different machine learning setups to highlight the nature of DG problems. The causality literature refers to DG as *learning from multiple environments* (Peters et al., 2016; Arjovsky et al., 2019). Although challenging, the DG framework can capture some of the difficulty of real prediction problems, where unforeseen distributional discrepancies between training and testing data are surely expected. At the same time, the framework can be limiting: in many real world scenarios there may be external variables informing about task relatedness (space, time, annotations) that the DG framework ignores.

## 3 MODEL SELECTION AS PART OF THE LEARNING PROBLEM

Here we discuss issues surrounding model selection (choosing hyperparameters, training checkpoints, architecture variants) in DG and make specific recommendations for a path forward. Because we lack access to a validation set identically distributed to the test data, model selection in DG is not as straightforward as in supervised learning. Some works adopt heuristic strategies whose behavior is not well-studied, while others simply omit a description of how to choose hyperparameters. This leaves open the possibility that hyperparameters were chosen using the test data, which is not

methodologically sound. Differences in results arising from inconsistent tuning practices may be misattributed to the algorithms under study, complicating fair assessments.

We believe that much of the confusion surrounding model selection in DG arises from treating it as merely a question of experimental design. To the contrary, model selection requires making theoretical assumptions about how the test data relates to the training data. Different DG algorithms make different assumptions, and it is not clear *a priori* which ones are correct, or how they influence the model selection criterion. Indeed, choosing reasonable assumptions is at the heart of DG research. Therefore, a DG algorithm without a strategy to choose its hyperparameters should be regarded as incomplete.

**Recommendation 1** *A DG algorithm should be responsible for specifying a model selection method.*

While algorithms without well-justified model selection methods are incomplete, they may be useful stepping-stones in a research agenda. In this case, instead of using an ad-hoc model selection method, we can evaluate incomplete algorithms by considering an *oracle model selection method*, where we select hyperparameters using some data from the test domain. Of course, it is important to avoid invalid comparisons between oracle results and baselines tuned without an oracle method. Also, unless we restrict access to the test domain data somehow, we risk obtaining meaningless results (we could just train on such test domain data using supervised learning).

**Recommendation 2** *Researchers should disclaim any oracle-selection results as such and specify policies to limit access to the test domain.*

## 3.1 THREE MODEL SELECTION METHODS FOR DG

Having made broad recommendations, we review and justify three model selection criteria for DG. Appendix B.3 illustrates these with an specific example.

**Training-domain validation**   We split each training domain into training and validation subsets. We train models using the training subsets, and choose the model maximizing the accuracy on the union of validation subsets. This strategy assumes that the training and test examples follow similar distributions. For example, Ben-David et al. (2010) bound the test error of a classifier with the divergence between training and test domains.

**Leave-one-domain-out validation**   Given $d_{\mathrm{tr}}$ training domains, we train $d_{\mathrm{tr}}$ models with equal hyperparameters, each holding one of the training domains out. We evaluate each model on its held-out domain, and average the accuracies of these $d_{\mathrm{tr}}$ models over their held-out domains. Finally, we choose the model maximizing this average accuracy, retrained on all $d_{\mathrm{tr}}$ domains. This strategy assumes that training and test domains follow a *meta-distribution* over domains, and that our goal is to maximize the expected performance under this meta-distribution. Note that leaving $k > 1$ domains out would increase greatly the number of experiments, and introduces a hyperparameter $k$.

**Test-domain validation (oracle)**   We choose the model maximizing the accuracy on a validation set that follows the distribution of the test domain. Following our earlier recommendation to limit test domain access, we allow one query (the last checkpoint) per choice of hyperparameters, disallowing early stopping. Recall that this is not a valid benchmarking methodology. Oracle-based results can be either optimistic, because we select models using the test distribution, or pessimistic, because the query limit reduces the number of considered hyperparameters. We also tried limiting the size of the oracle test set instead of the number of queries, but this led to unacceptably high variance.

## 3.2 CONSIDERATIONS FROM THE LITERATURE

Some references in prior work discuss additional strategies to choose hyperparameters in DG. For instance, Krueger et al. (2020, Appendix B.1) suggest choosing hyperparameters to maximize the performance across all domains of an external dataset. This "leave-one-dataset out" is akin to the second strategy outlined above. Albuquerque et al. (2019, Section 5.3.2) suggest performing model selection based on the loss function (which often incorporates an algorithm-specific regularizer), and D'Innocente and Caputo (2018, Section 3) derive an strategy specific to their algorithm. Finally, tools

from differential privacy enable multiple reuses of a validation set (Dwork et al., 2015), which could be a tool to control the power of test-domain validation (oracle).

## 4 DOMAINBED: A PYTORCH TESTBED FOR DOMAIN GENERALIZATION

At the heart of our large scale experimentation is DOMAINBED, a PyTorch (Paszke et al., 2019) testbed to streamline reproducible and rigorous research in DG:

https://github.com/facebookresearch/DomainBed/.

The initial release comprises fourteen algorithms, seven datasets, and three model selection methods (those described in Section 3), as well as the infrastructure to run all the experiments and generate all the LaTeX tables below with a single command. The first version of DOMAINBED focuses on image classification, leaving for future work other types of tasks. DOMAINBED is a living project: together with pull requests from collaborators, we continuously update the above repository with new algorithms, datasets, and result tables. As illustrated in Appendix B.5, adding a new algorithm or dataset to DOMAINBED is a matter of a few lines of code.

**Algorithms** DOMAINBED currently includes fourteen algorithms chosen based on their impact over the years, their published performance, and a desire to include varied DG strategies. These are Empirical Risk Minimization (**ERM**, Vapnik (1998)), Group Distributionally Robust Optimization (**GroupDRO**, Sagawa et al. (2019)), Inter-domain Mixup (**Mixup**, Xu et al. (2019); Yan et al. (2020); Wang et al. (2020b)), Meta-Learning for Domain Generalization (**MLDG**, Li et al. (2018a)), Domain-Adversarial Neural Networks (**DANN**, Ganin et al. (2016)), Class-conditional DANN (**C-DANN**, Li et al. (2018d)), Deep CORrelation ALignment (**CORAL**, Sun and Saenko (2016)), Maximum Mean Discrepancy (**MMD**, Li et al. (2018b)), Invariant Risk Minimization (**IRM** Arjovsky et al. (2019)), Adaptive Risk Minimization (**ARM**, Zhang et al. (2020)), Marginal Transfer Learning (**MTL**, Blanchard et al. (2011; 2017)), Style-Agnostic Networks (**SagNet**, Nam et al. (2019)), and Representation Self Challenging (**RSC**, Huang et al. (2020)). Appendix B.1 describes these algorithms, and Appendix B.4 lists their network architectures and hyperparameter search distributions.

**Datasets** DOMAINBED currently includes downloaders and loaders for seven standard DG image classification benchmarks. These are **Colored MNIST** (Arjovsky et al., 2019), **Rotated MNIST** (Ghifary et al., 2015), **PACS** (Li et al., 2017), **VLCS** (Fang et al., 2013), **OfficeHome** (Venkateswara et al., 2017), **Terra Incognita** (Beery et al., 2018), and **DomainNet** (Peng et al., 2019). The datasets based on MNIST are "synthetic" since changes across domains are well understood (colors and rotations). The rest of the datasets are "real" since domains vary in unknown ways. Appendix B.2 describes these datasets.

**Implementation choices** We highlight three implementation choices made towards a consistent and realistic evaluation setting. First, whereas prior work is inconsistent in its choice of network architecture, we finetune ResNet-50 models (He et al., 2016) pretrained on ImageNet for all non-MNIST experiments. We note that recent state-of-the-art results (Balaji et al., 2018; Nam et al., 2019; Huang et al., 2020) also use ResNet-50 models. Second, for all non-MNIST datasets, we augment training data using the following protocol: crops of random size and aspect ratio, resizing to $224 \times 224$ pixels, random horizontal flips, random color jitter, grayscaling the image with 10% probability, and normalization using the ImageNet channel statistics. This augmentation protocol is increasingly standard in state-of-the-art DG work (Nam et al., 2019; Huang et al., 2020; Krueger et al., 2020; Carlucci et al., 2019a; Zhou et al., 2020; Dou et al., 2019; Hendrycks et al., 2020; Wang et al., 2020a; Seo et al., 2020; Chattopadhyay et al., 2020). We use no augmentation for MNIST-based datasets. Third, and for RotatedMNIST, we divide all the digits evenly among domains, instead of replicating the same 1000 digits to construct all domains. We deviate from standard practice for two reasons: using the same digits across training and test domains leaks test data, and reducing the amount of training data complicates the task in an unrealistic way.

## 5 Experiments

We run experiments for all algorithms, datasets, and model selection criteria shipped in DOMAINBED. We consider all configurations of a dataset where we hide one domain for testing, resulting in the training of 58,000 models. To generate the following results, we simply run `sweep.py` at commit `0x7df6f06` from DOMAINBED's repository.

**Hyperparameter search**  For each algorithm and test domain, we conduct a random search (Bergstra and Bengio, 2012) of 20 trials over a joint distribution of all hyperparameters (Appendix B.4). Appendix C.4 shows that running more than 20 trials does not improve our results significantly. We use each model selection criterion to select amongst the 20 models from the random search. We split the data from each domain into 80% and 20% splits. We use the larger splits for training and final evaluation, and the smaller splits to select hyperparameters (for an illustration, see Appendix B.3). All hyperparameters are optimized anew for each algorithm and test domain, including hyperparameters like learning rates which are common to multiple algorithms.

**Standard error bars**  While some DG literature reports error bars across seeds, randomness arising from model selection is often ignored. This is acceptable if the goal is best-versus-best comparison, but prohibits analyses concerning the model selection process itself. Instead, we repeat *our entire study* three times, making every random choice anew: hyperparameters, weight initializations, and dataset splits. Every number we report is a mean (and its standard error) over these repetitions.

### 5.1 Results

Table 3 summarizes the results of our experiments. Appendix C contains the full results per dataset *and* domain. As anticipated in our introduction, we draw three conclusions from our results.

**Claim 1: Carefully tuned ERM outperforms the previously published state-of-the-art**  Table 1 (full version in Appendix A.5) shows this result, when we provide ERM with a training-domain validation set for hyperparameter selection. Such state-of-the-art average performance of our ERM baseline holds even when we select the best competitor available in the literature separately for each benchmark. One reason for ERM's strong performance is that we use ResNet-50, whereas some prior work uses smaller ResNet-18 models. As recently shown in the literature (Hendrycks et al., 2020), this suggests that better in-distribution generalization is a dominant factor behind better out-of-distribution generalization. Our result does not refute prior work: it is possible that with stronger implementations, some competing methods may improve upon ERM. Rather, we provide a strong, realistic, and reproducible baseline for future work to build upon.

**Claim 2: When evaluated in a consistent setting, no algorithm outperforms ERM in more than one point**  We observe this result in Table 3, obtained by running from scratch every combination of dataset, algorithm, and model selection criterion in DOMAINBED. Given any model selection criterion, no method improves the average performance of ERM in more than one point. At the number of trials performed, no improvement over ERM is statistically significant according to a $t$-test at a significance level $\alpha = 0.05$. While new algorithms could improve upon ERM (an exciting premise!), getting substantial DG improvements in a rigorous way proved challenging. Most of our baselines can achieve ERM-like performance because there have hyperparameter configurations under which they behave like ERM (e.g. regularization coefficients that can be set to zero). Our advice to DG practitioners is to use ERM (which is a safe contender) or CORAL (Sun and Saenko, 2016) (which achieved the highest average score).

**Claim 3: Model selection methods matter**  We observe that model selection with a training domain validation set outperforms leave-one-domain-out cross-validation across multiple datasets and algorithms. This does not mean that using a training domain validation set is *the right way* to tune hyperparameters. In fact, the stronger performance of oracle-selection ($+2.3$ points for ERM) suggests headroom to develop improved DG model selection criteria.

Table 3: DG accuracy for all algorithms, datasets and model selection criteria in DOMAINBED. These experiments compare fourteen popular DG algorithms across seven benchmarks in the exact same conditions, showing the competitive performance of ERM.

| Algorithm | CMNIST | RMNIST | VLCS | PACS | OfficeHome | TerraInc | DomainNet | Average |
|---|---|---|---|---|---|---|---|---|
| ERM | $51.5 \pm 0.1$ | $98.0 \pm 0.0$ | $77.5 \pm 0.4$ | $85.5 \pm 0.2$ | $66.5 \pm 0.3$ | $46.1 \pm 1.8$ | $40.9 \pm 0.1$ | 66.6 |
| IRM | $52.0 \pm 0.1$ | $97.7 \pm 0.1$ | $78.5 \pm 0.5$ | $83.5 \pm 0.8$ | $64.3 \pm 2.2$ | $47.6 \pm 0.8$ | $33.9 \pm 2.8$ | 65.4 |
| GroupDRO | $52.1 \pm 0.0$ | $98.0 \pm 0.0$ | $76.7 \pm 0.6$ | $84.4 \pm 0.8$ | $66.0 \pm 0.7$ | $43.2 \pm 1.1$ | $33.3 \pm 0.2$ | 64.8 |
| Mixup | $52.1 \pm 0.2$ | $98.0 \pm 0.1$ | $77.4 \pm 0.6$ | $84.6 \pm 0.6$ | $68.1 \pm 0.3$ | $47.9 \pm 0.8$ | $39.2 \pm 0.1$ | 66.7 |
| MLDG | $51.5 \pm 0.1$ | $97.9 \pm 0.0$ | $77.2 \pm 0.4$ | $84.9 \pm 1.0$ | $66.8 \pm 0.6$ | $47.7 \pm 0.9$ | $41.2 \pm 0.1$ | 66.7 |
| CORAL | $51.5 \pm 0.1$ | $98.0 \pm 0.1$ | $78.8 \pm 0.6$ | $86.2 \pm 0.3$ | $68.7 \pm 0.3$ | $47.6 \pm 1.0$ | $41.5 \pm 0.1$ | 67.5 |
| MMD | $51.5 \pm 0.2$ | $97.9 \pm 0.0$ | $77.5 \pm 0.9$ | $84.6 \pm 0.5$ | $66.3 \pm 0.1$ | $42.2 \pm 1.6$ | $23.4 \pm 9.5$ | 63.3 |
| DANN | $51.5 \pm 0.3$ | $97.8 \pm 0.1$ | $78.6 \pm 0.4$ | $83.6 \pm 0.4$ | $65.9 \pm 0.6$ | $46.7 \pm 0.5$ | $38.3 \pm 0.1$ | 66.1 |
| CDANN | $51.7 \pm 0.1$ | $97.9 \pm 0.1$ | $77.5 \pm 0.1$ | $82.6 \pm 0.9$ | $65.8 \pm 1.3$ | $45.8 \pm 1.6$ | $38.3 \pm 0.3$ | 65.6 |
| MTL | $51.4 \pm 0.1$ | $97.9 \pm 0.0$ | $77.2 \pm 0.4$ | $84.6 \pm 0.5$ | $66.4 \pm 0.5$ | $45.6 \pm 1.2$ | $40.6 \pm 0.1$ | 66.2 |
| SagNet | $51.7 \pm 0.0$ | $98.0 \pm 0.0$ | $77.8 \pm 0.5$ | $86.3 \pm 0.2$ | $68.1 \pm 0.1$ | $48.6 \pm 1.0$ | $40.3 \pm 0.1$ | 67.2 |
| ARM | $56.2 \pm 0.2$ | $98.2 \pm 0.1$ | $77.6 \pm 0.3$ | $85.1 \pm 0.4$ | $64.8 \pm 0.3$ | $45.5 \pm 0.3$ | $35.5 \pm 0.2$ | 66.1 |
| VREx | $51.8 \pm 0.1$ | $97.9 \pm 0.1$ | $78.3 \pm 0.2$ | $84.9 \pm 0.6$ | $66.4 \pm 0.6$ | $46.4 \pm 0.6$ | $33.6 \pm 2.9$ | 65.6 |
| RSC | $51.7 \pm 0.2$ | $97.6 \pm 0.1$ | $77.1 \pm 0.5$ | $85.2 \pm 0.9$ | $65.5 \pm 0.9$ | $46.6 \pm 1.0$ | $38.9 \pm 0.5$ | 66.1 |

Model selection: training-domain validation set

| Algorithm | CMNIST | RMNIST | VLCS | PACS | OfficeHome | TerraInc | DomainNet | Average |
|---|---|---|---|---|---|---|---|---|
| ERM | $36.7 \pm 0.1$ | $97.7 \pm 0.0$ | $77.2 \pm 0.4$ | $83.0 \pm 0.7$ | $65.7 \pm 0.5$ | $41.4 \pm 1.4$ | $40.6 \pm 0.2$ | 63.2 |
| IRM | $40.3 \pm 4.2$ | $97.0 \pm 0.2$ | $76.3 \pm 0.6$ | $81.5 \pm 0.8$ | $64.3 \pm 1.5$ | $41.2 \pm 3.6$ | $33.5 \pm 3.0$ | 62.0 |
| GroupDRO | $36.8 \pm 0.1$ | $97.6 \pm 0.1$ | $77.9 \pm 0.5$ | $83.5 \pm 0.2$ | $65.2 \pm 0.2$ | $44.9 \pm 1.4$ | $33.0 \pm 0.3$ | 62.7 |
| Mixup | $33.4 \pm 4.7$ | $97.8 \pm 0.0$ | $77.7 \pm 0.6$ | $83.2 \pm 0.4$ | $67.0 \pm 0.2$ | $48.7 \pm 0.4$ | $38.5 \pm 0.3$ | 63.8 |
| MLDG | $36.7 \pm 0.2$ | $97.6 \pm 0.0$ | $77.2 \pm 0.9$ | $82.9 \pm 1.7$ | $66.1 \pm 0.5$ | $46.2 \pm 0.9$ | $41.0 \pm 0.2$ | 64.0 |
| CORAL | $39.7 \pm 2.8$ | $97.8 \pm 0.1$ | $78.7 \pm 0.4$ | $82.6 \pm 0.5$ | $68.5 \pm 0.2$ | $46.3 \pm 1.7$ | $41.1 \pm 0.1$ | 65.0 |
| MMD | $36.8 \pm 0.1$ | $97.8 \pm 0.1$ | $77.3 \pm 0.5$ | $83.2 \pm 0.2$ | $60.2 \pm 5.2$ | $46.5 \pm 1.5$ | $23.4 \pm 9.5$ | 60.7 |
| DANN | $40.7 \pm 2.3$ | $97.6 \pm 0.2$ | $76.9 \pm 0.4$ | $81.0 \pm 1.1$ | $64.9 \pm 1.2$ | $44.4 \pm 1.1$ | $38.2 \pm 0.2$ | 63.4 |
| CDANN | $39.1 \pm 4.4$ | $97.5 \pm 0.2$ | $77.5 \pm 0.2$ | $78.8 \pm 2.2$ | $64.3 \pm 1.7$ | $39.9 \pm 3.2$ | $38.0 \pm 0.1$ | 62.2 |
| MTL | $35.0 \pm 1.7$ | $97.8 \pm 0.1$ | $76.6 \pm 0.5$ | $83.7 \pm 0.4$ | $65.7 \pm 0.5$ | $44.9 \pm 1.2$ | $40.6 \pm 0.1$ | 63.5 |
| SagNet | $36.5 \pm 0.1$ | $94.0 \pm 3.0$ | $77.5 \pm 0.3$ | $82.3 \pm 0.1$ | $67.6 \pm 0.3$ | $47.2 \pm 0.9$ | $40.2 \pm 0.2$ | 63.6 |
| ARM | $36.8 \pm 0.0$ | $98.1 \pm 0.1$ | $76.6 \pm 0.5$ | $81.7 \pm 0.2$ | $64.4 \pm 0.2$ | $42.6 \pm 2.7$ | $35.2 \pm 0.1$ | 62.2 |
| VREx | $36.9 \pm 0.3$ | $93.6 \pm 3.4$ | $76.7 \pm 1.0$ | $81.3 \pm 0.9$ | $64.9 \pm 1.3$ | $37.3 \pm 3.0$ | $33.4 \pm 3.1$ | 60.6 |
| RSC | $36.5 \pm 0.2$ | $97.6 \pm 0.1$ | $77.5 \pm 0.5$ | $82.6 \pm 0.7$ | $65.8 \pm 0.7$ | $40.0 \pm 0.8$ | $38.9 \pm 0.5$ | 62.7 |

Model selection: leave-one-domain-out cross-validation

| Algorithm | CMNIST | RMNIST | VLCS | PACS | OfficeHome | TerraInc | DomainNet | Average |
|---|---|---|---|---|---|---|---|---|
| ERM | $57.8 \pm 0.2$ | $97.8 \pm 0.1$ | $77.6 \pm 0.3$ | $86.7 \pm 0.3$ | $66.4 \pm 0.5$ | $53.0 \pm 0.3$ | $41.3 \pm 0.1$ | 68.7 |
| IRM | $67.7 \pm 1.2$ | $97.5 \pm 0.2$ | $76.9 \pm 0.6$ | $84.5 \pm 1.1$ | $63.0 \pm 2.7$ | $50.5 \pm 0.7$ | $28.0 \pm 5.1$ | 66.9 |
| GroupDRO | $61.1 \pm 0.9$ | $97.9 \pm 0.1$ | $77.4 \pm 0.6$ | $87.1 \pm 0.1$ | $66.2 \pm 0.6$ | $52.4 \pm 0.1$ | $33.4 \pm 0.3$ | 67.9 |
| Mixup | $58.4 \pm 0.2$ | $98.0 \pm 0.1$ | $78.1 \pm 0.3$ | $86.8 \pm 0.3$ | $68.0 \pm 0.2$ | $54.4 \pm 0.3$ | $39.6 \pm 0.1$ | 69.0 |
| MLDG | $58.2 \pm 0.4$ | $97.8 \pm 0.1$ | $77.5 \pm 0.1$ | $86.8 \pm 0.4$ | $66.6 \pm 0.3$ | $52.0 \pm 0.1$ | $41.6 \pm 0.1$ | 68.7 |
| CORAL | $58.6 \pm 0.5$ | $98.0 \pm 0.0$ | $77.7 \pm 0.2$ | $87.1 \pm 0.5$ | $68.4 \pm 0.2$ | $52.8 \pm 0.2$ | $41.8 \pm 0.1$ | 69.2 |
| MMD | $63.3 \pm 1.3$ | $98.0 \pm 0.1$ | $77.9 \pm 0.1$ | $87.2 \pm 0.1$ | $66.2 \pm 0.3$ | $52.0 \pm 0.4$ | $23.5 \pm 9.4$ | 66.9 |
| DANN | $57.0 \pm 1.0$ | $97.9 \pm 0.1$ | $79.7 \pm 0.5$ | $85.2 \pm 0.2$ | $65.3 \pm 0.8$ | $50.6 \pm 0.4$ | $38.3 \pm 0.1$ | 67.7 |
| CDANN | $59.5 \pm 2.0$ | $97.9 \pm 0.0$ | $79.9 \pm 0.2$ | $85.8 \pm 0.8$ | $65.3 \pm 0.5$ | $50.8 \pm 0.6$ | $38.5 \pm 0.2$ | 68.2 |
| MTL | $57.6 \pm 0.3$ | $97.9 \pm 0.1$ | $77.7 \pm 0.5$ | $86.7 \pm 0.2$ | $66.5 \pm 0.4$ | $52.2 \pm 0.4$ | $40.8 \pm 0.1$ | 68.5 |
| SagNet | $58.2 \pm 0.3$ | $97.9 \pm 0.0$ | $77.6 \pm 0.1$ | $86.4 \pm 0.4$ | $67.5 \pm 0.2$ | $52.5 \pm 0.4$ | $40.8 \pm 0.2$ | 68.7 |
| ARM | $63.2 \pm 0.7$ | $98.1 \pm 0.1$ | $77.8 \pm 0.3$ | $85.8 \pm 0.2$ | $64.8 \pm 0.4$ | $51.2 \pm 0.5$ | $36.0 \pm 0.2$ | 68.1 |
| VREx | $67.0 \pm 1.3$ | $97.9 \pm 0.1$ | $78.1 \pm 0.2$ | $87.2 \pm 0.6$ | $65.7 \pm 0.3$ | $51.4 \pm 0.5$ | $30.1 \pm 3.7$ | 68.2 |
| RSC | $58.5 \pm 0.5$ | $97.6 \pm 0.1$ | $77.8 \pm 0.6$ | $86.2 \pm 0.5$ | $66.5 \pm 0.6$ | $52.1 \pm 0.2$ | $38.9 \pm 0.6$ | 68.2 |

Model selection: test-domain validation set (oracle)

Table 4: Ablation study on ERM showing the impact of (i) using raw images versus data augmentation, and (ii) using ResNet-18 versus ResNet-50 models. Model selection: training-domain validation set.

| Algorithm | VLCS | PACS | OfficeHome | TerraInc | DomainNet | Avg |
|---|---|---|---|---|---|---|
| ERM (raw, 18) | $75.8 \pm 0.3$ | $79.6 \pm 0.3$ | $61.0 \pm 0.1$ | $35.0 \pm 1.3$ | $35.8 \pm 0.2$ | 62.4 |
| ERM (aug, 18) | $75.8 \pm 0.1$ | $79.1 \pm 0.8$ | $60.0 \pm 0.6$ | $40.0 \pm 0.6$ | $35.3 \pm 0.0$ | 62.8 |
| ERM (raw, 50) | $78.6 \pm 0.1$ | $83.2 \pm 0.6$ | $67.7 \pm 0.2$ | $41.5 \pm 2.5$ | $41.4 \pm 0.1$ | 66.0 |
| ERM (aug, 50) | $77.5 \pm 0.4$ | $85.5 \pm 0.2$ | $66.5 \pm 0.3$ | $46.1 \pm 1.8$ | $40.9 \pm 0.1$ | 66.6 |

## 5.2 ABLATION STUDY ON ERM

To better understand our ERM performance, we perform an ablation study on the neural network architecture and the data augmentation protocol. Table 5.2 shows that using a ResNet-50 neural network architecture, instead of a smaller ResNet-18, improves DG test accuracy by 3.7 points. Using data augmentation improves DG test accuracy by 0.5 points. However, these ResNet models were pretrained on ImageNet *using data augmentation*, so the benefits of augmentation are partly absorbed by the model. In fact, we hypothesize that among models pretrained on ImageNet, domain generalization performance is mainly influenced by the model's original test accuracy on ImageNet.

## 6 DISCUSSIONS

We provide several discussions to help the reader interpret our results and motivate future work.

**Our negative claims are fundamentally limited** Broad negative claims (e.g. "algorithm X does not outperform ERM") do not specify an exact experimental setting and are therefore impossible to rigorously prove. In order to be verifiable, such claims must be restricted to a specific setting. This limitation is fundamental to all negative result claims, and ours (Claim 2) is no exception. We have shown that many algorithms don't substantially improve on ERM *in our setting*, but the relevance of that setting is a subjective matter ultimately left for the reader.

In making this judgement, the reader should consider whether they agree with our methodological and implementation choices, which we have explained and motivated throughout the paper. We also note that our implementation can outperform previous results (Table 1). Finally, DomainBed is not a black box: our implementation is open-source and actively maintained, and we invite the research community to improve on our results.

**Is this as good as it gets?** We question whether DG is possible in some of the considered datasets. Why do we assume that a neural network should be able to classify cartoons, given only photorealistic training data? In the case of Rotated MNIST, do truly rotation-invariant features discriminative of the digit class exist? Are those features expressible by a neural network? Even in the presence of correct model selection, is the out-of-distribution performance of modern ERM implementations as good as it gets? Or is it simply as poor as every other alternative? How far are we from the achievable DG performance? Is this upper-bound simply the test error in-domain?

**Are these the right datasets?** Most datasets considered in the DG literature do not reflect realistic situations. If one wanted to classify cartoons, the easiest option would be to collect a small labeled dataset of cartoons. Should we consider more realistic, impactful tasks for better research in DG? Some alternatives are medical imaging in different hospitals and self-driving cars in different cities.

**Generalization requires untestable assumptions** Every time we use ERM, we assume that training and testing examples follow the same distribution. This is an untestable assumption in every single instance. The same applies for DG: each algorithm assumes a different (untestable) type of invariance across domains. Therefore, the performance of a DG algorithm depends on the problem at hand, and only time can tell if we have made a good choice. This is akin to the generalization of a scientific theory such as Newton's gravitation, which cannot be proven, but rather only resist falsification.

## 7 CONCLUSION

Our extensive empirical evaluation of DG algorithms leads to three conclusions. First, a carefully tuned ERM baseline outperforms the previously published state-of-the-art results in terms of average performance (**Claim 1**). Second, when compared to thirteen popular DG alternatives on the exact same experimental conditions, we find out that no competitor is able to outperform ERM by more than one point (**Claim 2**). Third, model selection is non-trivial for DG, and it should be an integral part of any proposed method (**Claim 3**). Going forward, we hope that our results and DOMAINBED promote realistic and rigorous evaluation and enable advances in domain generalization.

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

## A A DECADE OF LITERATURE ON DOMAIN GENERALIZATION

In this section, we provide an exhaustive literature review on a decade of domain generalization research. The following classifies domain generalization algorithms according into four strategies to learn invariant predictors: learning invariant features, sharing parameters, meta-learning, or performing data augmentation.

### A.1 LEARNING INVARIANT FEATURES

Muandet et al. (2013) use kernel methods to find a feature transformation that (i) minimizes the distance between transformed feature distributions across domains, and (ii) does not destroy any of the information between the original features and the targets. In their pioneering work, Ganin et al. (2016) propose Domain Adversarial Neural Networks (DANN), a domain adaptation technique which uses generative adversarial networks (GANs, Goodfellow et al. (2014)), to learn a feature representation that matches across training domains. Akuzawa et al. (2019) extend DANN by considering cases where there exists an statistical dependence between the domain and the class label variables. Albuquerque et al. (2019) extend DANN by considering one-versus-all adversaries that try to predict to which training domain does each of the examples belong to. Li et al. (2018b) employ GANs and the maximum mean discrepancy criteria (Gretton et al., 2012) to align feature distributions across domains. Matsuura and Harada (2019) leverages clustering techniques to learn domain-invariant features even when the separation between training domains is not given. Li et al. (2018c;d) learns a feature transformation $\phi$ such that the conditional distributions $P(\phi(X^d) \mid Y^d = y)$ match for all training domains $d$ and label values $y$. Shankar et al. (2018) use a domain classifier to construct adversarial examples for a label classifier, and use a label classifier to construct adversarial examples for the domain classifier. This results in a label classifier with better domain generalization. Li et al. (2019a) train a robust feature extractor and classifier. The robustness comes from (i) asking the feature extractor to produce features such that a classifier trained on domain $d$ can classify instances for domain $d' \neq d$, and (ii) asking the classifier to predict labels on domain $d$ using features produced by a feature extractor trained on domain $d' \neq d$. Li et al. (2020) adopt a lifelong learning strategy to attack the problem of domain generalization. Motiian et al. (2017) learn a feature representation such that (i) examples from different domains but the same class are close, (ii) examples from different domains and classes are far, and (iii) training examples can be correctly classified. Ilse et al. (2019) train a variational autoencoder (Kingma and Welling, 2014) where the bottleneck representation factorizes knowledge about domain, class label, and residual variations in the input space. Fang et al. (2013) learn a structural SVM metric such that the neighborhood of each example contains examples from the same category and all training domains. The algorithms of Sun and Saenko (2016); Sun et al. (2016); Rahman et al. (2019a) match the feature covariance (second order statistics) across training domains at some level of representation. The algorithms of Ghifary et al. (2016); Hu et al. (2019) use kernel-based multivariate component analysis to minimize the mismatch between training domains while maximizing class separability.

Although popular, learning domain-invariant features has received some criticism (Zhao et al., 2019; Johansson et al., 2019). Some alternatives exist, as we review next. Peters et al. (2016); Rojas-Carulla et al. (2018) considered that one should search for features that lead to the same optimal classifier across training domains. In their pioneering work, Peters et al. (2016) linked this type of invariance to the causal structure of data, and provided a basic algorithm to learn invariant linear models, based on feature selection. Arjovsky et al. (2019) extend the previous to general gradient-based models, including neural networks, in their Invariant Risk Minimization (IRM) principle. Teney et al. (2020) build on IRM to learn a feature transformation that minimizes the relative variance of classifier weights across training datasets. The authors apply their method to reduce the learning of spurious correlations in Visual Question Answering (VQA) tasks. Ahuja et al. (2020) analyze IRM under a game-theoretic perspective to develop an alternative algorithm. Krueger et al. (2020) propose an approximation to the IRM problem consisting in reducing the variance of error averages across domains. Bouvier et al. (2019) attack the same problem as IRM by re-weighting data samples.

### A.2 SHARING PARAMETERS

Blanchard et al. (2011) build classifiers $f(x^d, \mu^d)$, where $\mu^d$ is a kernel mean embedding (Muandet et al., 2017) that summarizes the dataset associated to the example $x^d$. Since the distributional

identity of test instances is unknown, these embeddings are estimated using single test examples at test time. See Blanchard et al. (2017); Deshmukh et al. (2019) for theoretical results on this family of algorithms (only applicable when using RKHS-based learners). Zhang et al. (2020) is an extension of Blanchard et al. (2011) where a separate CNN computes the domain embedding, appended to the input image as additional channels. Khosla et al. (2012) learn one max-margin linear classifier $w^d = w + \Delta^d$ per domain $d$, from which they distill their final, invariant predictor $w$. Ghifary et al. (2015) use a multitask autoencoder to learn invariances across domains. To achieve this, the authors assume that each training dataset contains the same examples; for instance, photographs about the same objects under different views. Mancini et al. (2018b) train a deep neural network with one set of dedicated batch-normalization layers (Ioffe and Szegedy, 2015) per training dataset. Then, a softmax domain classifier predicts how to linearly-combine the batch-normalization layers at test time. Seo et al. (2020) combines instance normalization with batch-normalization to learn a normalization module per domain, enhancing out-of-distribution generalization. Similarly, Mancini et al. (2018a) learn a softmax domain classifier used to linearly-combine domain-specific predictors at test time. D'Innocente and Caputo (2018) explore more sophisticated ways of aggregating domain-specific predictors. Li et al. (2017) extends Khosla et al. (2012) to deep neural networks by extending each of their parameter tensors with one additional dimension, indexed by the training domains, and set to a neutral value to predict domain-agnostic test examples. Ding and Fu (2017) implement parameter-tying and low-rank reconstruction losses to learn a predictor that relies on common knowledge across training domains. Hu et al. (2016); Sagawa et al. (2019) weight the importance of the minibatches of the training distributions proportional to their error. Chattopadhyay et al. (2020) overlays multiple weight masks over a single network to learn domain-invariant and domain-specific features.

## A.3 META-LEARNING

Li et al. (2018a) employ Model-Agnostic Meta-Learning, or MAML (Finn et al., 2017), to build a predictor that learns how to adapt fast between training domains. Dou et al. (2019) use a similar MAML strategy, together with two regularizers that encourage features from different domains to respect inter-class relationships, and be compactly clustered by class labels. Li et al. (2019b) extend the MAML meta-learning strategy to instances of domain generalization where the categories vary from domain to domain. Balaji et al. (2018) use MAML to meta-learn a regularizer encouraging the model trained on one domain to perform well on another domain.

## A.4 AUGMENTING DATA

Data augmentation is an effective strategy to address domain generalization (Zhang et al., 2019). Unfortunately, how to design efficient data augmentation routines depends on the type of data at hand, and demands a significant amount of work from human experts. Xu et al. (2019); Yan et al. (2020); Wang et al. (2020b) use mixup (Zhang et al., 2018) to blend examples from the different training distributions. Carlucci et al. (2019a) constructs an auxiliary classification task aimed at solving jigsaw puzzles of image patches. The authors show that this self-supervised learning task learns features that improve domain generalization. Similarly, Wang et al. (2020a) use metric learning and self-supervised learning to augment the out-of-distribution performance of an image classifier. Albuquerque et al. (2020) introduce the self-supervised task of predicting responses to Gabor filter banks, in order to learn more transferrable features. Wang et al. (2019) remove textural information from images to improve domain generalization. Volpi et al. (2018) show that training with adversarial data augmentation on a single domain is sufficient to improve domain generalization. Nam et al. (2019) promote representations of data that ignore image style and focus on content. Rahman et al. (2019b); Zhou et al. (2020); Carlucci et al. (2019a) are three alternatives that use GANs to augment the data available during training time. Representation Self-Challenging (Huang et al., 2020) learns robust neural networks by iteratively dropping-out important features. Hendrycks et al. (2020) show that, together with larger models and data, data augmentation improves out-of-distribution performance.

## A.5 PREVIOUS STATE-OF-THE-ART NUMBERS

Table 5 compiles the best out-of-distribution test accuracies reported across a decade of domain generalization research.

Table 5: Previous state-of-the-art in the literature of domain generalization.

| Benchmark | Accuracy (by domain) | | | | | | | Algorithm |
|---|---|---|---|---|---|---|---|---|
| | 0 | 15 | 30 | 45 | 60 | 75 | Average | |
| | 82.50 | 96.30 | 93.40 | 78.60 | 94.20 | 80.50 | 87.58 | D-MTAE (Ghifary et al., 2015) |
| | 84.60 | 95.60 | 94.60 | 82.90 | 94.80 | 82.10 | 89.10 | CCSA (Motiian et al., 2017) |
| | 83.70 | 96.90 | 95.70 | 85.20 | 95.90 | 81.20 | 89.80 | MMD-AAE (Li et al., 2018b) |
| | 85.60 | 95.00 | 95.60 | 95.50 | 95.90 | 84.30 | 92.00 | BestSources (Mancini et al., 2018a) |
| Rotated | 88.80 | 97.60 | 97.50 | 97.80 | 97.60 | 91.90 | 95.20 | ADAGE (Carlucci et al., 2019b) |
| MNIST | 88.30 | 98.60 | 98.00 | 97.70 | 97.70 | 91.40 | 95.28 | CrossGrad (Shankar et al., 2018) |
| | 90.10 | 98.90 | 98.90 | 98.80 | 98.30 | 90.00 | 95.80 | HEX (Wang et al., 2019) |
| | 89.23 | 99.68 | 99.20 | 99.24 | 99.53 | 91.44 | 96.39 | FeatureCritic (Li et al., 2019b) |
| | 93.50 | 99.30 | 99.10 | 99.20 | 99.30 | 93.00 | 97.20 | DIVA (Ilse et al., 2019) |
| | 95.90 | 98.90 | 98.80 | 98.90 | 98.90 | 96.40 | 98.00 | **Our ERM** |
| | C | L | S | V | | | Average | |
| | 88.92 | 59.60 | 59.20 | 64.36 | | | 64.06 | SCA (Ghifary et al., 2016) |
| | 92.30 | 62.10 | 59.10 | 67.10 | | | 65.00 | CCSA (Motiian et al., 2017) |
| | 89.15 | 64.99 | 58.88 | 62.59 | | | 67.67 | MTSSL (Albuquerque et al., 2020) |
| | 89.05 | 60.13 | 61.33 | 63.90 | | | 68.60 | D-MTAE (Ghifary et al., 2015) |
| | 91.12 | 60.43 | 60.85 | 65.65 | | | 69.41 | CIDG (Li et al., 2018c) |
| | 88.83 | 63.06 | 62.10 | 64.38 | | | 69.59 | CIDDG (Li et al., 2018d) |
| | 92.64 | 61.78 | 59.60 | 66.86 | | | 70.22 | MDA (Hu et al., 2019) |
| | 92.76 | 62.34 | 63.54 | 65.25 | | | 70.97 | MDA (Ding and Fu, 2017) |
| | 93.63 | 63.49 | 61.32 | 69.99 | | | 72.11 | DBADG (Li et al., 2017) |
| | 94.40 | 62.60 | 64.40 | 67.60 | | | 72.30 | MMD-AAE (Li et al., 2018b) |
| VLCS | 94.10 | 64.30 | 65.90 | 67.10 | | | 72.90 | Epi-FCR (Li et al., 2019a) |
| | 96.93 | 60.90 | 64.30 | 70.62 | | | 73.19 | JiGen (Carlucci et al., 2019a) |
| | 96.72 | 60.40 | 63.68 | 70.49 | | | 73.30 | VREx (Krueger et al., 2020) |
| | 96.40 | 64.80 | 64.00 | 68.70 | | | 73.50 | S-MLDG (Li et al., 2020) |
| | 96.66 | 58.77 | 68.13 | 71.96 | | | 73.88 | MMLD (Matsuura and Harada, 2019) |
| | 94.78 | 64.90 | 67.64 | 69.14 | | | 74.11 | MASF (Dou et al., 2019) |
| | 97.33 | 63.49 | 68.02 | 69.83 | | | 74.67 | EISNet (Wang et al., 2020a) |
| | 97.61 | 61.86 | 68.32 | 73.93 | | | 75.43 | RSC (Huang et al., 2020) |
| | 95.52 | 67.63 | 69.37 | 71.14 | | | 75.92 | G2DM (Albuquerque et al., 2019) |
| | 97.70 | 64.30 | 73.40 | 74.60 | | | 77.50 | **Our ERM** |
| | A | C | P | S | | | Average | |
| | 62.86 | 66.97 | 89.50 | 57.51 | | | 69.21 | DBADG (Li et al., 2017) |
| | 61.67 | 67.41 | 84.31 | 63.91 | | | 69.32 | MTSSL (Albuquerque et al., 2020) |
| | 62.70 | 69.73 | 78.65 | 64.45 | | | 69.40 | CIDDG (Li et al., 2018d) |
| | 62.64 | 65.98 | 90.44 | 58.76 | | | 69.45 | JAN-COMBO (Rahman et al., 2019b) |
| | 66.23 | 66.88 | 88.00 | 58.96 | | | 70.01 | MLDG (Li et al., 2018a) |
| | 66.80 | 69.70 | 87.90 | 56.30 | | | 70.20 | HEX (Wang et al., 2019) |
| | 64.10 | 66.80 | 90.20 | 60.10 | | | 70.30 | BestSources (Mancini et al., 2018a) |
| | 64.40 | 68.60 | 90.10 | 58.40 | | | 70.40 | FeatureCritic (Li et al., 2019b) |
| | 67.04 | 67.97 | 89.74 | 59.81 | | | 71.14 | VREx (Krueger et al., 2020) |
| | 65.52 | 69.90 | 89.16 | 63.37 | | | 71.98 | CAADG (Rahman et al., 2019a) |
| | 64.70 | 72.30 | 86.10 | 65.00 | | | 72.00 | Epi-FCR (Li et al., 2019a) |
| | 66.60 | 73.36 | 88.12 | 66.19 | | | 73.55 | G2DM (Albuquerque et al., 2019) |
| PACS | 70.35 | 72.46 | 90.68 | 67.33 | | | 75.21 | MASF (Dou et al., 2019) |
| | 79.42 | 75.25 | 96.03 | 71.35 | | | 80.51 | JiGen (Carlucci et al., 2019a) |
| | 80.50 | 77.80 | 94.80 | 72.80 | | | 81.50 | S-MLDG (Li et al., 2020) |
| | 79.48 | 77.13 | 94.30 | 75.30 | | | 81.55 | D-SAM-Λ (D'Innocente and Caputo, 2018) |
| | 81.28 | 77.16 | 96.09 | 72.29 | | | 81.83 | MMLD (Matsuura and Harada, 2019) |
| | 84.20 | 78.10 | 95.30 | 74.70 | | | 83.10 | DDAIG (Zhou et al., 2020) |
| | 83.58 | 77.66 | 95.47 | 76.30 | | | 83.25 | SagNets (Nam et al., 2019) |
| | 82.57 | 78.11 | 94.49 | 78.32 | | | 83.37 | DMG (Chattopadhyay et al., 2020) |
| | 87.20 | 79.20 | 97.60 | 70.30 | | | 83.60 | MetaReg (Balaji et al., 2018) |
| | 84.70 | 80.80 | 97.20 | 79.30 | | | 85.50 | **Our ERM** |
| | 86.64 | 81.53 | 97.11 | 78.07 | | | 85.84 | EISNet (Wang et al., 2020a) |
| | 87.04 | 80.62 | 95.99 | 82.90 | | | 86.64 | DSON (Seo et al., 2020) |
| | 87.89 | 82.16 | 97.92 | 83.35 | | | 87.83 | RSC (Huang et al., 2020) |
| | A | C | P | R | | | Average | |
| | 48.09 | 45.20 | 66.52 | 68.35 | | | 57.04 | JAN-COMBO (Rahman et al., 2019b) |
| | 53.04 | 47.51 | 71.47 | 72.79 | | | 61.20 | JiGen (Carlucci et al., 2019a) |
| Office | 54.53 | 49.04 | 71.57 | 71.90 | | | 61.76 | D-SAM-Λ (D'Innocente and Caputo, 2018) |
| Home | 60.20 | 45.38 | 70.42 | 73.38 | | | 62.34 | SagNets (Nam et al., 2019) |
| | 59.37 | 45.70 | 71.84 | 74.68 | | | 62.90 | DSON (Seo et al., 2020) |
| | 58.42 | 47.90 | 71.63 | 74.54 | | | 63.12 | RSC (Huang et al., 2020) |
| | 59.20 | 52.30 | 74.60 | 76.00 | | | 65.50 | DDAIG (Zhou et al., 2020) |
| | 61.30 | 52.40 | 75.80 | 76.60 | | | 66.50 | **Our ERM** |

## B  MORE ABOUT DOMAINBED

### B.1  ALGORITHMS

1. Empirical Risk Minimization (**ERM**, Vapnik (1998)) minimizes the errors across domains.
2. Group Distributionally Robust Optimization (**DRO**, Sagawa et al. (2019)) performs ERM while increasing the importance of domains with larger errors.
3. Inter-domain Mixup (**Mixup**, Xu et al. (2019); Yan et al. (2020); Wang et al. (2020b)) performs ERM on linear interpolations of examples from random pairs of domains and their labels.
4. Meta-Learning for Domain Generalization (**MLDG**, Li et al. (2018a)) leverages MAML (Finn et al., 2017) to meta-learn how to generalize across domains.
5. Domain-Adversarial Neural Networks (**DANN**, Ganin et al. (2016)) employ an adversarial network to match feature distributions across environments.
6. Class-conditional DANN (**C-DANN**, Li et al. (2018d)) is a variant of DANN matching the conditional distributions $P(\phi(X^d)|Y^d = y)$ across domains, for all labels $y$.
7. **CORAL** (Sun and Saenko, 2016) matches the mean and covariance of feature distributions.
8. **MMD** (Li et al., 2018b) matches the MMD (Gretton et al., 2012) of feature distributions.
9. Invariant Risk Minimization (**IRM** Arjovsky et al. (2019)) learns a feature representation $\phi(X^d)$ such that the optimal linear classifier on top of that representation matches across domains.
10. Risk Extrapolation (**VREx**, Krueger et al. (2020)) approximates IRM with a variance penalty.
11. Marginal Transfer Learning (**MTL**, Blanchard et al. (2011; 2017)) estimates a mean embedding per domain, passed as a second argument to the classifier.
12. Adaptive Risk Min. (**ARM**, Zhang et al. (2020)) extends **MTL** with a separate embedding CNN.
13. Style-Agnostic Networks (**SagNets**, Nam et al. (2019)) learns neural networks by keeping image content and randomizing style.
14. Representation Self-Challenging (**RSC**, Huang et al. (2020)) learns robust neural networks by iteratively discarding (challenging) the most activated features.

### B.2  DATASETS

DOMAINBED includes downloaders and loaders for seven multi-domain image classification tasks:

1. **Colored MNIST** (Arjovsky et al., 2019) is a variant of the MNIST handwritten digit classification dataset (LeCun, 1998). Domain $d \in \{0.1, 0.3, 0.9\}$ contains a disjoint set of digits colored either red or blue. The label is a noisy function of the digit and color, such that color bears correlation $d$ with the label and the digit bears correlation 0.75 with the label. This dataset contains $70,000$ examples of dimension $(2, 28, 28)$ and 2 classes.
2. **Rotated MNIST** (Ghifary et al., 2015) is a variant of MNIST where domain $d \in \{$ 0, 15, 30, 45, 60, 75 $\}$ contains digits rotated by $d$ degrees. Our dataset contains $70,000$ examples of dimension $(1, 28, 28)$ and 10 classes.
3. **PACS** (Li et al., 2017) comprises four domains $d \in \{$ art, cartoons, photos, sketches $\}$. This dataset contains $9,991$ examples of dimension $(3, 224, 224)$ and 7 classes.
4. **VLCS** (Fang et al., 2013) comprises photographic domains $d \in \{$ Caltech101, LabelMe, SUN09, VOC2007 $\}$. This dataset contains $10,729$ examples of dimension $(3, 224, 224)$ and 5 classes.
5. **OfficeHome** (Venkateswara et al., 2017) includes domains $d \in \{$ art, clipart, product, real $\}$. This dataset contains $15,588$ examples of dimension $(3, 224, 224)$ and 65 classes.
6. **Terra Incognita** (Beery et al., 2018) contains photographs of wild animals taken by camera traps at locations $d \in \{L100, L38, L43, L46\}$. Our version of this dataset contains $24,788$ examples of dimension $(3, 224, 224)$ and 10 classes.
7. **DomainNet** (Peng et al., 2019) has six domains $d \in \{$ clipart, infograph, painting, quickdraw, real, sketch $\}$. This dataset contains $586,575$ examples of size $(3, 224, 224)$ and 345 classes.

For all datasets, we first pool the raw training, validation, and testing images together. For each random seed, we then instantiate random training, validation, and testing splits.

### B.3 Model selection criteria, illustrated

Consider Figure 1, and let $T_i = \{A_i, B_i, C_i\}$ for $i \in \{1, 2\}$. *Training-domain validation* trains each hyperparameter configuration on $T_1$ and chooses the configuration with the highest performance in $T_2$. *Leave-one-out validation* trains one clone $F_Z$ of each hyperparameter configuration on $T_1 \setminus Z$, for $Z \in T_1$; then, it chooses the configuration with highest $\sum_{Z \in T_1} \text{Performance}(F_Z, Z)$. *Test-domain validation* trains each hyperparameter configuration on $T_1$ and chooses the configuration with the highest performance on $D_2$, only looking at its final epoch. Finally, result tables show the performance of selected models on $D_1$.

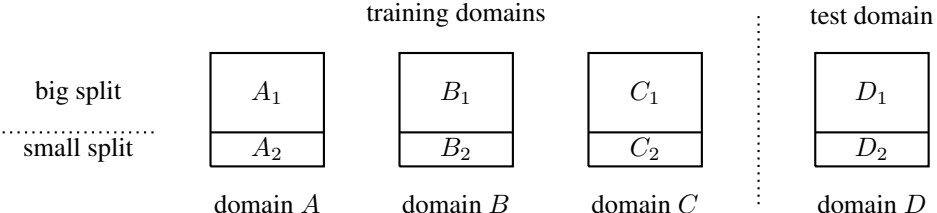

Figure 1: Data configuration for a benchmark with four domains A, B, C, D, where the test domain is D. We shuffle and divide the data from each domain into a big split and a small split.

### B.4 Architectures and hyperparameters

Neural network architectures used for each dataset:

| Dataset | Architecture |
|---|---|
| Colored MNIST
Rotated MNIST | MNIST ConvNet |
| PACS
VLCS
OfficeHome
TerraIncognita | ResNet-50 |

Neural network architecture for MNIST experiments:

| # | Layer |
|---|---|
| 1 | Conv2D (in=$d$, out=64) |
| 2 | ReLU |
| 3 | GroupNorm (groups=8) |
| 4 | Conv2D (in=64, out=128, stride=2) |
| 5 | ReLU |
| 6 | GroupNorm (8 groups) |
| 7 | Conv2D (in=128, out=128) |
| 8 | ReLU |
| 9 | GroupNorm (8 groups) |
| 10 | Conv2D (in=128, out=128) |
| 11 | ReLU |
| 12 | GroupNorm (8 groups) |
| 13 | Global average-pooling |

For "ResNet-50", we replace the final (softmax) layer of a ResNet50 pretrained on ImageNet and fine-tune the entire network. Since minibatches from different domains follow different distributions, batch normalization degrades domain generalization algorithms (Seo et al., 2020). Therefore, we freeze all batch normalization layers before fine-tuning. We insert a dropout layer before the final ResNet-50 linear layer.

Table 6 lists all algorithm hyperparameters, their default values, and their sweep random search distribution. We optimize all models using Adam (Kingma and Ba, 2015).

Table 6: Hyperparameters, their default values and distributions for random search.

| Condition | Parameter | Default value | Random distribution |
|---|---|---|---|
| ResNet | learning rate | 0.00005 | $10^{\text{Uniform}(-5,-3.5)}$ |
| | batch size | 32 | $2^{\text{Uniform}(3,5.5)}$ |
| | batch size (if ARM) | 8 | 8 |
| | ResNet dropout | 0 | $\text{RandomChoice}([0, 0.1, 0.5])$ |
| | generator learning rate | 0.00005 | $10^{\text{Uniform}(-5,-3.5)}$ |
| | discriminator learning rate | 0.00005 | $10^{\text{Uniform}(-5,-3.5)}$ |
| not ResNet | learning rate | 0.001 | $10^{\text{Uniform}(-4.5,-3.5)}$ |
| | batch size | 64 | $2^{\text{Uniform}(3,9)}$ |
| | generator learning rate | 0.001 | $10^{\text{Uniform}(-4.5,-2.5)}$ |
| | discriminator learning rate | 0.001 | $10^{\text{Uniform}(-4.5,-2.5)}$ |
| MNIST | weight decay | 0 | 0 |
| | generator weight decay | 0 | 0 |
| not MNIST | weight decay | 0 | $10^{\text{Uniform}(-6,-2)}$ |
| | generator weight decay | 0 | $10^{\text{Uniform}(-6,-2)}$ |
| DANN, C-DANN | lambda | 1.0 | $10^{\text{Uniform}(-2,2)}$ |
| | discriminator weight decay | 0 | $10^{\text{Uniform}(-6,-2)}$ |
| | discriminator steps | 1 | $2^{\text{Uniform}(0,3)}$ |
| | discriminator width | 256 | $\text{int}(2^{\text{Uniform}(6,10)})$ |
| | discriminator depth | 3 | $\text{RandomChoice}([3, 4, 5])$ |
| | discriminator dropout | 0 | $\text{RandomChoice}([0, 0.1, 0.5])$ |
| | discriminator grad penalty | 0 | $10^{\text{Uniform}(-2,1)}$ |
| | Adam $\beta_1$ | 0.5 | $\text{RandomChoice}([0, 0.5])$ |
| DRO | eta | 0.01 | $10^{\text{Uniform}(-1,1)}$ |
| IRM | lambda | 100 | $10^{\text{Uniform}(-1,5)}$ |
| | warmup iterations | 500 | $10^{\text{Uniform}(0,4)}$ |
| Mixup | alpha | 0.2 | $10^{\text{Uniform}(0,4)}$ |
| MLDG | beta | 1 | $10^{\text{Uniform}(-1,1)}$ |
| MMD | gamma | 1 | $10^{\text{Uniform}(-1,1)}$ |
| MTL | ema | 0.99 | $\text{RandomChoice}([.5, .9, .99, 1])$ |
| RSC | feature drop percentage | 1/3 | $\text{Uniform}(0, 0.5)$ |
| | batch drop percentage | 1/3 | $\text{Uniform}(0, 0.5)$ |
| SagNet | adversary weight | 0.1 | $10^{\text{Uniform}(-2,1)}$ |
| VREx | lambda | 10 | $10^{\text{Uniform}(-1,5)}$ |
| | warmup iterations | 500 | $10^{\text{Uniform}(0,4)}$ |

## B.5 EXTENDING DOMAINBED

Algorithms are classes that implement two methods: `.update(minibatches)` and `.predict(x)`. The update method receives a list of minibatches, one minibatch per training domain, and each minibatch containing one input and one output tensor. For example, to implement group DRO (Sagawa et al., 2019, Algorithm 1), we simply write the following in `algorithms.py`:

```python
class GroupDRO(ERM):
    def __init__(self, input_shape, num_classes, num_domains, hparams):
        super().__init__(input_shape, num_classes, num_domains, hparams)
        self.register_buffer("q", torch.Tensor())

    def update(self, minibatches):
        device = "cuda" if minibatches[0][0].is_cuda else "cpu"

        if not len(self.q):
            self.q = torch.ones(len(minibatches)).to(device)

        losses = torch.zeros(len(minibatches)).to(device)

        for m in range(len(minibatches)):
            x, y = minibatches[m]
            losses[m] = F.cross_entropy(self.predict(x), y)
            self.q[m] *= (self.hparams["dro_eta"] * losses[m].data).exp()

        self.q /= self.q.sum()
        loss = torch.dot(losses, self.q) / len(minibatches)

        self.optimizer.zero_grad()
        loss.backward()
        self.optimizer.step()

        return {'loss': loss.item()}

ALGORITHMS.append('GroupDRO')
```

By inheriting from ERM, the new `GroupDRO` class has access to a default classifier `.network`, optimizer `.optimizer`, and prediction method `.predict(x)`. Finally, we tell DOMAINBED about the default values and hyperparameter search distributions of the hyperparameters of this new algorithm. We do so by adding the following to the function `_hparams` in `hparams_registry.py`:

```python
hparams['dro_eta'] = (1e-2, 10**random_state.uniform(-3, -1))
```

To add a new image classification dataset to DOMAINBED, arrange your image files as `/root/MyDataset/domain/class/image.jpg`. Then, append to `datasets.py`:

```python
class MyDataset(MultipleEnvironmentImageFolder):
    ENVIRONMENTS = ['Env1', 'Env2', 'Env3']
    def __init__(self, root, test_envs, augment=True):
        self.dir = os.path.join(root, "MyDataset/")
        super().__init__(self.dir, test_envs, augment)

DATASETS.append('MyDataset')
```

We are now ready to train our new algorithm on our new dataset, using the second domain as test:

```
python train.py --model DRO --dataset MyDataset --data_dir /root --test_envs 1 \
                --hparams '{"dro_eta": 0.2}'
```

Finally, we can run a fully automated sweep on all datasets, algorithms, test domains, and model selection criteria by simply invoking `python sweep.py`, after extending the file `command_launchers.py` to your computing infrastructure. When the sweep finishes, the script `collect_results.py` automatically generates all the result tables shown in this manuscript.

**Extension to UDA** One can use DOMAINBED to perform experimentation on unsupervised domain adaptation by extending the `.update(minibatches)` methods to accept unlabeled examples from the test domain.

# C    FULL DOMAINBED RESULTS

## C.1    MODEL SELECTION: TRAINING-DOMAIN VALIDATION SET

### C.1.1    COLOREDMNIST

| Algorithm | +90% | +80% | -90% | Avg |
|---|---|---|---|---|
| ERM | 71.7 ± 0.1 | 72.9 ± 0.2 | 10.0 ± 0.1 | 51.5 |
| IRM | 72.5 ± 0.1 | 73.3 ± 0.5 | 10.2 ± 0.3 | 52.0 |
| GroupDRO | 73.1 ± 0.3 | 73.2 ± 0.2 | 10.0 ± 0.2 | 52.1 |
| Mixup | 72.7 ± 0.4 | 73.4 ± 0.1 | 10.1 ± 0.1 | 52.1 |
| MLDG | 71.5 ± 0.2 | 73.1 ± 0.2 | 9.8 ± 0.1 | 51.5 |
| CORAL | 71.6 ± 0.3 | 73.1 ± 0.1 | 9.9 ± 0.1 | 51.5 |
| MMD | 71.4 ± 0.3 | 73.1 ± 0.2 | 9.9 ± 0.3 | 51.5 |
| DANN | 71.4 ± 0.9 | 73.1 ± 0.1 | 10.0 ± 0.0 | 51.5 |
| CDANN | 72.0 ± 0.2 | 73.0 ± 0.2 | 10.2 ± 0.1 | 51.7 |
| MTL | 70.9 ± 0.2 | 72.8 ± 0.3 | 10.5 ± 0.1 | 51.4 |
| SagNet | 71.8 ± 0.2 | 73.0 ± 0.2 | 10.3 ± 0.0 | 51.7 |
| ARM | 82.0 ± 0.5 | 76.5 ± 0.3 | 10.2 ± 0.0 | 56.2 |
| VREx | 72.4 ± 0.3 | 72.9 ± 0.4 | 10.2 ± 0.0 | 51.8 |
| RSC | 71.9 ± 0.3 | 73.1 ± 0.2 | 10.0 ± 0.2 | 51.7 |

### C.1.2    ROTATEDMNIST

| Algorithm | 0 | 15 | 30 | 45 | 60 | 75 | Avg |
|---|---|---|---|---|---|---|---|
| ERM | 95.9 ± 0.1 | 98.9 ± 0.0 | 98.8 ± 0.0 | 98.9 ± 0.0 | 98.9 ± 0.0 | 96.4 ± 0.0 | 98.0 |
| IRM | 95.5 ± 0.1 | 98.8 ± 0.2 | 98.7 ± 0.1 | 98.6 ± 0.1 | 98.7 ± 0.0 | 95.9 ± 0.2 | 97.7 |
| GroupDRO | 95.6 ± 0.1 | 98.9 ± 0.1 | 98.9 ± 0.1 | 99.0 ± 0.0 | 98.9 ± 0.0 | 96.5 ± 0.2 | 98.0 |
| Mixup | 95.8 ± 0.3 | 98.9 ± 0.0 | 98.9 ± 0.0 | 98.9 ± 0.0 | 98.8 ± 0.1 | 96.5 ± 0.3 | 98.0 |
| MLDG | 95.8 ± 0.1 | 98.9 ± 0.1 | 99.0 ± 0.0 | 98.9 ± 0.1 | 99.0 ± 0.0 | 95.8 ± 0.3 | 97.9 |
| CORAL | 95.8 ± 0.3 | 98.8 ± 0.0 | 98.9 ± 0.0 | 99.0 ± 0.0 | 98.9 ± 0.1 | 96.4 ± 0.2 | 98.0 |
| MMD | 95.6 ± 0.1 | 98.9 ± 0.1 | 99.0 ± 0.0 | 99.0 ± 0.0 | 98.9 ± 0.0 | 96.0 ± 0.2 | 97.9 |
| DANN | 95.0 ± 0.5 | 98.9 ± 0.1 | 99.0 ± 0.0 | 99.0 ± 0.1 | 98.9 ± 0.0 | 96.3 ± 0.2 | 97.8 |
| CDANN | 95.7 ± 0.2 | 98.8 ± 0.0 | 98.9 ± 0.1 | 98.9 ± 0.1 | 98.9 ± 0.1 | 96.1 ± 0.3 | 97.9 |
| MTL | 95.6 ± 0.1 | 99.0 ± 0.1 | 99.0 ± 0.0 | 98.9 ± 0.1 | 99.0 ± 0.1 | 95.8 ± 0.2 | 97.9 |
| SagNet | 95.9 ± 0.3 | 98.9 ± 0.1 | 99.0 ± 0.1 | 99.1 ± 0.0 | 99.0 ± 0.1 | 96.3 ± 0.1 | 98.0 |
| ARM | 96.7 ± 0.2 | 99.1 ± 0.0 | 99.0 ± 0.0 | 99.0 ± 0.1 | 99.1 ± 0.1 | 96.5 ± 0.4 | 98.2 |
| VREx | 95.9 ± 0.2 | 99.0 ± 0.1 | 98.9 ± 0.1 | 98.9 ± 0.1 | 98.7 ± 0.1 | 96.2 ± 0.2 | 97.9 |
| RSC | 94.8 ± 0.5 | 98.7 ± 0.1 | 98.8 ± 0.1 | 98.8 ± 0.0 | 98.9 ± 0.1 | 95.9 ± 0.2 | 97.6 |

### C.1.3    VLCS

| Algorithm | C | L | S | V | Avg |
|---|---|---|---|---|---|
| ERM | 97.7 ± 0.4 | 64.3 ± 0.9 | 73.4 ± 0.5 | 74.6 ± 1.3 | 77.5 |
| IRM | 98.6 ± 0.1 | 64.9 ± 0.9 | 73.4 ± 0.6 | 77.3 ± 0.9 | 78.5 |
| GroupDRO | 97.3 ± 0.3 | 63.4 ± 0.9 | 69.5 ± 0.8 | 76.7 ± 0.7 | 76.7 |
| Mixup | 98.3 ± 0.6 | 64.8 ± 1.0 | 72.1 ± 0.5 | 74.3 ± 0.8 | 77.4 |
| MLDG | 97.4 ± 0.2 | 65.2 ± 0.7 | 71.0 ± 1.4 | 75.3 ± 1.0 | 77.2 |
| CORAL | 98.3 ± 0.1 | 66.1 ± 1.2 | 73.4 ± 0.3 | 77.5 ± 1.2 | 78.8 |
| MMD | 97.7 ± 0.1 | 64.0 ± 1.1 | 72.8 ± 0.2 | 75.3 ± 3.3 | 77.5 |
| DANN | 99.0 ± 0.3 | 65.1 ± 1.4 | 73.1 ± 0.3 | 77.2 ± 0.6 | 78.6 |
| CDANN | 97.1 ± 0.3 | 65.1 ± 1.2 | 70.7 ± 0.8 | 77.1 ± 1.5 | 77.5 |
| MTL | 97.8 ± 0.4 | 64.3 ± 0.3 | 71.5 ± 0.7 | 75.3 ± 1.7 | 77.2 |
| SagNet | 97.9 ± 0.4 | 64.5 ± 0.5 | 71.4 ± 1.3 | 77.5 ± 0.5 | 77.8 |
| ARM | 98.7 ± 0.2 | 63.6 ± 0.7 | 71.3 ± 1.2 | 76.7 ± 0.6 | 77.6 |
| VREx | 98.4 ± 0.3 | 64.4 ± 1.4 | 74.1 ± 0.4 | 76.2 ± 1.3 | 78.3 |
| RSC | 97.9 ± 0.1 | 62.5 ± 0.7 | 72.3 ± 1.2 | 75.6 ± 0.8 | 77.1 |

### C.1.4   PACS

| Algorithm | A | C | P | S | Avg |
|---|---|---|---|---|---|
| ERM | $84.7 \pm 0.4$ | $80.8 \pm 0.6$ | $97.2 \pm 0.3$ | $79.3 \pm 1.0$ | 85.5 |
| IRM | $84.8 \pm 1.3$ | $76.4 \pm 1.1$ | $96.7 \pm 0.6$ | $76.1 \pm 1.0$ | 83.5 |
| GroupDRO | $83.5 \pm 0.9$ | $79.1 \pm 0.6$ | $96.7 \pm 0.3$ | $78.3 \pm 2.0$ | 84.4 |
| Mixup | $86.1 \pm 0.5$ | $78.9 \pm 0.8$ | $97.6 \pm 0.1$ | $75.8 \pm 1.8$ | 84.6 |
| MLDG | $85.5 \pm 1.4$ | $80.1 \pm 1.7$ | $97.4 \pm 0.3$ | $76.6 \pm 1.1$ | 84.9 |
| CORAL | $88.3 \pm 0.2$ | $80.0 \pm 0.5$ | $97.5 \pm 0.3$ | $78.8 \pm 1.3$ | 86.2 |
| MMD | $86.1 \pm 1.4$ | $79.4 \pm 0.9$ | $96.6 \pm 0.2$ | $76.5 \pm 0.5$ | 84.6 |
| DANN | $86.4 \pm 0.8$ | $77.4 \pm 0.8$ | $97.3 \pm 0.4$ | $73.5 \pm 2.3$ | 83.6 |
| CDANN | $84.6 \pm 1.8$ | $75.5 \pm 0.9$ | $96.8 \pm 0.3$ | $73.5 \pm 0.6$ | 82.6 |
| MTL | $87.5 \pm 0.8$ | $77.1 \pm 0.5$ | $96.4 \pm 0.8$ | $77.3 \pm 1.8$ | 84.6 |
| SagNet | $87.4 \pm 1.0$ | $80.7 \pm 0.6$ | $97.1 \pm 0.1$ | $80.0 \pm 0.4$ | 86.3 |
| ARM | $86.8 \pm 0.6$ | $76.8 \pm 0.5$ | $97.4 \pm 0.3$ | $79.3 \pm 1.2$ | 85.1 |
| VREx | $86.0 \pm 1.6$ | $79.1 \pm 0.6$ | $96.9 \pm 0.5$ | $77.7 \pm 1.7$ | 84.9 |
| RSC | $85.4 \pm 0.8$ | $79.7 \pm 1.8$ | $97.6 \pm 0.3$ | $78.2 \pm 1.2$ | 85.2 |

### C.1.5   OFFICEHOME

| Algorithm | A | C | P | R | Avg |
|---|---|---|---|---|---|
| ERM | $61.3 \pm 0.7$ | $52.4 \pm 0.3$ | $75.8 \pm 0.1$ | $76.6 \pm 0.3$ | 66.5 |
| IRM | $58.9 \pm 2.3$ | $52.2 \pm 1.6$ | $72.1 \pm 2.9$ | $74.0 \pm 2.5$ | 64.3 |
| GroupDRO | $60.4 \pm 0.7$ | $52.7 \pm 1.0$ | $75.0 \pm 0.7$ | $76.0 \pm 0.7$ | 66.0 |
| Mixup | $62.4 \pm 0.8$ | $54.8 \pm 0.6$ | $76.9 \pm 0.3$ | $78.3 \pm 0.2$ | 68.1 |
| MLDG | $61.5 \pm 0.9$ | $53.2 \pm 0.6$ | $75.0 \pm 1.2$ | $77.5 \pm 0.4$ | 66.8 |
| CORAL | $65.3 \pm 0.4$ | $54.4 \pm 0.5$ | $76.5 \pm 0.1$ | $78.4 \pm 0.5$ | 68.7 |
| MMD | $60.4 \pm 0.2$ | $53.3 \pm 0.3$ | $74.3 \pm 0.1$ | $77.4 \pm 0.6$ | 66.3 |
| DANN | $59.9 \pm 1.3$ | $53.0 \pm 0.3$ | $73.6 \pm 0.7$ | $76.9 \pm 0.5$ | 65.9 |
| CDANN | $61.5 \pm 1.4$ | $50.4 \pm 2.4$ | $74.4 \pm 0.9$ | $76.6 \pm 0.8$ | 65.8 |
| MTL | $61.5 \pm 0.7$ | $52.4 \pm 0.6$ | $74.9 \pm 0.4$ | $76.8 \pm 0.4$ | 66.4 |
| SagNet | $63.4 \pm 0.2$ | $54.8 \pm 0.4$ | $75.8 \pm 0.4$ | $78.3 \pm 0.3$ | 68.1 |
| ARM | $58.9 \pm 0.8$ | $51.0 \pm 0.5$ | $74.1 \pm 0.1$ | $75.2 \pm 0.3$ | 64.8 |
| VREx | $60.7 \pm 0.9$ | $53.0 \pm 0.9$ | $75.3 \pm 0.1$ | $76.6 \pm 0.5$ | 66.4 |
| RSC | $60.7 \pm 1.4$ | $51.4 \pm 0.3$ | $74.8 \pm 1.1$ | $75.1 \pm 1.3$ | 65.5 |

### C.1.6   TERRAINCOGNITA

| Algorithm | L100 | L38 | L43 | L46 | Avg |
|---|---|---|---|---|---|
| ERM | $49.8 \pm 4.4$ | $42.1 \pm 1.4$ | $56.9 \pm 1.8$ | $35.7 \pm 3.9$ | 46.1 |
| IRM | $54.6 \pm 1.3$ | $39.8 \pm 1.9$ | $56.2 \pm 1.8$ | $39.6 \pm 0.8$ | 47.6 |
| GroupDRO | $41.2 \pm 0.7$ | $38.6 \pm 2.1$ | $56.7 \pm 0.9$ | $36.4 \pm 2.1$ | 43.2 |
| Mixup | $59.6 \pm 2.0$ | $42.2 \pm 1.4$ | $55.9 \pm 0.8$ | $33.9 \pm 1.4$ | 47.9 |
| MLDG | $54.2 \pm 3.0$ | $44.3 \pm 1.1$ | $55.6 \pm 0.3$ | $36.9 \pm 2.2$ | 47.7 |
| CORAL | $51.6 \pm 2.4$ | $42.2 \pm 1.0$ | $57.0 \pm 1.0$ | $39.8 \pm 2.9$ | 47.6 |
| MMD | $41.9 \pm 3.0$ | $34.8 \pm 1.0$ | $57.0 \pm 1.9$ | $35.2 \pm 1.8$ | 42.2 |
| DANN | $51.1 \pm 3.5$ | $40.6 \pm 0.6$ | $57.4 \pm 0.5$ | $37.7 \pm 1.8$ | 46.7 |
| CDANN | $47.0 \pm 1.9$ | $41.3 \pm 4.8$ | $54.9 \pm 1.7$ | $39.8 \pm 2.3$ | 45.8 |
| MTL | $49.3 \pm 1.2$ | $39.6 \pm 6.3$ | $55.6 \pm 1.1$ | $37.8 \pm 0.8$ | 45.6 |
| SagNet | $53.0 \pm 2.9$ | $43.0 \pm 2.5$ | $57.9 \pm 0.6$ | $40.4 \pm 1.3$ | 48.6 |
| ARM | $49.3 \pm 0.7$ | $38.3 \pm 2.4$ | $55.8 \pm 0.8$ | $38.7 \pm 1.3$ | 45.5 |
| VREx | $48.2 \pm 4.3$ | $41.7 \pm 1.3$ | $56.8 \pm 0.8$ | $38.7 \pm 3.1$ | 46.4 |
| RSC | $50.2 \pm 2.2$ | $39.2 \pm 1.4$ | $56.3 \pm 1.4$ | $40.8 \pm 0.6$ | 46.6 |

### C.1.7 DOMAINNET

| Algorithm | clip | info | paint | quick | real | sketch | Avg |
|---|---|---|---|---|---|---|---|
| ERM | 58.1 ± 0.3 | 18.8 ± 0.3 | 46.7 ± 0.3 | 12.2 ± 0.4 | 59.6 ± 0.1 | 49.8 ± 0.4 | 40.9 |
| IRM | 48.5 ± 2.8 | 15.0 ± 1.5 | 38.3 ± 4.3 | 10.9 ± 0.5 | 48.2 ± 5.2 | 42.3 ± 3.1 | 33.9 |
| GroupDRO | 47.2 ± 0.5 | 17.5 ± 0.4 | 33.8 ± 0.5 | 9.3 ± 0.3 | 51.6 ± 0.4 | 40.1 ± 0.6 | 33.3 |
| Mixup | 55.7 ± 0.3 | 18.5 ± 0.5 | 44.3 ± 0.5 | 12.5 ± 0.4 | 55.8 ± 0.3 | 48.2 ± 0.5 | 39.2 |
| MLDG | 59.1 ± 0.2 | 19.1 ± 0.3 | 45.8 ± 0.7 | 13.4 ± 0.3 | 59.6 ± 0.2 | 50.2 ± 0.4 | 41.2 |
| CORAL | 59.2 ± 0.1 | 19.7 ± 0.2 | 46.6 ± 0.3 | 13.4 ± 0.4 | 59.8 ± 0.2 | 50.1 ± 0.6 | 41.5 |
| MMD | 32.1 ± 13.3 | 11.0 ± 4.6 | 26.8 ± 11.3 | 8.7 ± 2.1 | 32.7 ± 13.8 | 28.9 ± 11.9 | 23.4 |
| DANN | 53.1 ± 0.2 | 18.3 ± 0.1 | 44.2 ± 0.7 | 11.8 ± 0.1 | 55.5 ± 0.4 | 46.8 ± 0.6 | 38.3 |
| CDANN | 54.6 ± 0.4 | 17.3 ± 0.1 | 43.7 ± 0.9 | 12.1 ± 0.7 | 56.2 ± 0.4 | 45.9 ± 0.5 | 38.3 |
| MTL | 57.9 ± 0.5 | 18.5 ± 0.4 | 46.0 ± 0.1 | 12.5 ± 0.1 | 59.5 ± 0.3 | 49.2 ± 0.1 | 40.6 |
| SagNet | 57.7 ± 0.3 | 19.0 ± 0.2 | 45.3 ± 0.3 | 12.7 ± 0.5 | 58.1 ± 0.5 | 48.8 ± 0.2 | 40.3 |
| ARM | 49.7 ± 0.3 | 16.3 ± 0.5 | 40.9 ± 1.1 | 9.4 ± 0.1 | 53.4 ± 0.4 | 43.5 ± 0.4 | 35.5 |
| VREx | 47.3 ± 3.5 | 16.0 ± 1.5 | 35.8 ± 4.6 | 10.9 ± 0.3 | 49.6 ± 4.9 | 42.0 ± 3.0 | 33.6 |
| RSC | 55.0 ± 1.2 | 18.3 ± 0.5 | 44.4 ± 0.6 | 12.2 ± 0.2 | 55.7 ± 0.7 | 47.8 ± 0.9 | 38.9 |

### C.1.8 AVERAGES

| Algorithm | ColoredMNIST | RotatedMNIST | VLCS | PACS | OfficeHome | TerraIncognita | DomainNet | Avg |
|---|---|---|---|---|---|---|---|---|
| ERM | 51.5 ± 0.1 | 98.0 ± 0.0 | 77.5 ± 0.4 | 85.5 ± 0.2 | 66.5 ± 0.3 | 46.1 ± 1.8 | 40.9 ± 0.1 | 66.6 |
| IRM | 52.0 ± 0.1 | 97.7 ± 0.1 | 78.5 ± 0.5 | 83.5 ± 0.8 | 64.3 ± 2.2 | 47.6 ± 0.8 | 33.9 ± 2.8 | 65.4 |
| GroupDRO | 52.1 ± 0.0 | 98.0 ± 0.0 | 76.7 ± 0.6 | 84.4 ± 0.8 | 66.0 ± 0.7 | 43.2 ± 1.1 | 33.3 ± 0.2 | 64.8 |
| Mixup | 52.1 ± 0.2 | 98.0 ± 0.1 | 77.4 ± 0.6 | 84.6 ± 0.6 | 68.1 ± 0.3 | 47.9 ± 0.8 | 39.2 ± 0.1 | 66.7 |
| MLDG | 51.5 ± 0.1 | 97.9 ± 0.0 | 77.2 ± 0.4 | 84.9 ± 1.0 | 66.8 ± 0.6 | 47.7 ± 0.9 | 41.2 ± 0.1 | 66.7 |
| CORAL | 51.5 ± 0.1 | 98.0 ± 0.1 | 78.8 ± 0.6 | 86.2 ± 0.3 | 68.7 ± 0.3 | 47.6 ± 1.0 | 41.5 ± 0.1 | 67.5 |
| MMD | 51.5 ± 0.2 | 97.9 ± 0.0 | 77.5 ± 0.9 | 84.6 ± 0.5 | 66.3 ± 0.1 | 42.2 ± 1.6 | 23.4 ± 9.5 | 63.3 |
| DANN | 51.5 ± 0.3 | 97.8 ± 0.1 | 78.6 ± 0.4 | 83.6 ± 0.4 | 65.9 ± 0.6 | 46.7 ± 0.5 | 38.3 ± 0.1 | 66.1 |
| CDANN | 51.7 ± 0.1 | 97.9 ± 0.1 | 77.5 ± 0.1 | 82.6 ± 0.9 | 65.8 ± 1.3 | 45.8 ± 1.6 | 38.3 ± 0.3 | 65.6 |
| MTL | 51.4 ± 0.1 | 97.9 ± 0.0 | 77.2 ± 0.4 | 84.6 ± 0.5 | 66.4 ± 0.5 | 45.6 ± 1.2 | 40.6 ± 0.1 | 66.2 |
| SagNet | 51.7 ± 0.0 | 98.0 ± 0.0 | 77.8 ± 0.5 | 86.3 ± 0.2 | 68.1 ± 0.1 | 48.6 ± 1.0 | 40.3 ± 0.1 | 67.2 |
| ARM | 56.2 ± 0.2 | 98.2 ± 0.1 | 77.6 ± 0.3 | 85.1 ± 0.4 | 64.8 ± 0.3 | 45.5 ± 0.3 | 35.5 ± 0.2 | 66.1 |
| VREx | 51.8 ± 0.1 | 97.9 ± 0.1 | 78.3 ± 0.2 | 84.9 ± 0.6 | 66.4 ± 0.6 | 46.4 ± 0.6 | 33.6 ± 2.9 | 65.6 |
| RSC | 51.7 ± 0.2 | 97.6 ± 0.1 | 77.1 ± 0.5 | 85.2 ± 0.9 | 65.5 ± 0.9 | 46.6 ± 1.0 | 38.9 ± 0.5 | 66.1 |

### C.2 MODEL SELECTION: LEAVE-ONE-DOMAIN-OUT CROSS-VALIDATION

### C.2.1 COLOREDMNIST

| Algorithm | +90% | +80% | -90% | Avg |
|---|---|---|---|---|
| ERM | 50.0 ± 0.2 | 50.1 ± 0.2 | 10.0 ± 0.0 | 36.7 |
| IRM | 46.7 ± 2.4 | 51.2 ± 0.3 | 23.1 ± 10.7 | 40.3 |
| GroupDRO | 50.1 ± 0.5 | 50.0 ± 0.5 | 10.2 ± 0.1 | 36.8 |
| Mixup | 36.6 ± 10.9 | 53.4 ± 5.9 | 10.2 ± 0.1 | 33.4 |
| MLDG | 50.1 ± 0.6 | 50.1 ± 0.3 | 10.0 ± 0.1 | 36.7 |
| CORAL | 49.5 ± 0.0 | 59.5 ± 8.2 | 10.2 ± 0.1 | 39.7 |
| MMD | 50.3 ± 0.2 | 50.0 ± 0.4 | 9.9 ± 0.2 | 36.8 |
| DANN | 49.9 ± 0.1 | 62.1 ± 7.0 | 10.0 ± 0.1 | 40.7 |
| CDANN | 63.2 ± 10.1 | 44.4 ± 4.5 | 9.9 ± 0.2 | 39.1 |
| MTL | 44.3 ± 4.9 | 50.7 ± 0.0 | 10.1 ± 0.1 | 35.0 |
| SagNet | 49.9 ± 0.4 | 49.7 ± 0.3 | 10.0 ± 0.1 | 36.5 |
| ARM | 50.0 ± 0.3 | 50.1 ± 0.3 | 10.2 ± 0.0 | 36.8 |
| VREx | 50.2 ± 0.4 | 50.5 ± 0.5 | 10.1 ± 0.0 | 36.9 |
| RSC | 49.6 ± 0.3 | 49.7 ± 0.4 | 10.1 ± 0.0 | 36.5 |

### C.2.2 ROTATEDMNIST

| Algorithm | 0 | 15 | 30 | 45 | 60 | 75 | Avg |
|---|---|---|---|---|---|---|---|
| ERM | $95.3 \pm 0.2$ | $98.9 \pm 0.1$ | $98.9 \pm 0.1$ | $98.8 \pm 0.1$ | $98.5 \pm 0.1$ | $96.2 \pm 0.2$ | 97.7 |
| IRM | $94.5 \pm 0.5$ | $98.2 \pm 0.2$ | $98.7 \pm 0.1$ | $96.6 \pm 1.5$ | $98.4 \pm 0.1$ | $95.8 \pm 0.1$ | 97.0 |
| GroupDRO | $95.7 \pm 0.3$ | $98.7 \pm 0.1$ | $98.9 \pm 0.1$ | $98.6 \pm 0.2$ | $98.6 \pm 0.2$ | $95.3 \pm 0.9$ | 97.6 |
| Mixup | $94.8 \pm 0.4$ | $98.8 \pm 0.0$ | $98.9 \pm 0.1$ | $99.0 \pm 0.1$ | $98.9 \pm 0.0$ | $96.4 \pm 0.3$ | 97.8 |
| MLDG | $94.3 \pm 0.4$ | $98.8 \pm 0.1$ | $99.0 \pm 0.1$ | $98.8 \pm 0.1$ | $98.8 \pm 0.1$ | $96.0 \pm 0.3$ | 97.6 |
| CORAL | $95.7 \pm 0.5$ | $98.5 \pm 0.2$ | $98.9 \pm 0.2$ | $98.6 \pm 0.2$ | $98.8 \pm 0.1$ | $96.3 \pm 0.2$ | 97.8 |
| MMD | $95.8 \pm 0.2$ | $98.7 \pm 0.1$ | $99.0 \pm 0.0$ | $98.8 \pm 0.1$ | $98.7 \pm 0.1$ | $96.1 \pm 0.2$ | 97.8 |
| DANN | $95.1 \pm 0.5$ | $98.3 \pm 0.5$ | $98.5 \pm 0.1$ | $99.0 \pm 0.1$ | $98.6 \pm 0.1$ | $96.1 \pm 0.3$ | 97.6 |
| CDANN | $94.3 \pm 0.5$ | $98.4 \pm 0.3$ | $98.9 \pm 0.1$ | $98.7 \pm 0.1$ | $98.9 \pm 0.1$ | $95.7 \pm 0.4$ | 97.5 |
| MTL | $95.5 \pm 0.3$ | $98.6 \pm 0.3$ | $98.8 \pm 0.1$ | $99.0 \pm 0.1$ | $99.0 \pm 0.1$ | $95.6 \pm 0.3$ | 97.8 |
| SagNet | $94.0 \pm 1.6$ | $98.7 \pm 0.2$ | $98.9 \pm 0.1$ | $99.1 \pm 0.0$ | $98.8 \pm 0.1$ | $74.2 \pm 16.5$ | 94.0 |
| ARM | $95.8 \pm 0.1$ | $99.0 \pm 0.1$ | $99.0 \pm 0.0$ | $98.9 \pm 0.1$ | $98.8 \pm 0.1$ | $96.9 \pm 0.3$ | 98.1 |
| VREx | $95.8 \pm 0.2$ | $98.7 \pm 0.0$ | $98.5 \pm 0.1$ | $98.9 \pm 0.1$ | $74.0 \pm 20.1$ | $95.5 \pm 0.5$ | 93.6 |
| RSC | $94.6 \pm 0.0$ | $98.4 \pm 0.2$ | $99.0 \pm 0.1$ | $98.9 \pm 0.0$ | $98.8 \pm 0.1$ | $95.9 \pm 0.4$ | 97.6 |

### C.2.3 VLCS

| Algorithm | C | L | S | V | Avg |
|---|---|---|---|---|---|
| ERM | $98.0 \pm 0.4$ | $62.6 \pm 0.9$ | $70.8 \pm 1.9$ | $77.5 \pm 1.9$ | 77.2 |
| IRM | $98.6 \pm 0.3$ | $66.0 \pm 1.1$ | $69.3 \pm 0.9$ | $71.5 \pm 1.9$ | 76.3 |
| GroupDRO | $98.1 \pm 0.3$ | $66.4 \pm 0.9$ | $71.0 \pm 0.3$ | $76.1 \pm 1.4$ | 77.9 |
| Mixup | $98.4 \pm 0.3$ | $63.4 \pm 0.7$ | $72.9 \pm 0.8$ | $76.1 \pm 1.2$ | 77.7 |
| MLDG | $98.5 \pm 0.3$ | $61.7 \pm 1.2$ | $73.6 \pm 1.8$ | $75.0 \pm 0.8$ | 77.2 |
| CORAL | $96.9 \pm 0.9$ | $65.7 \pm 1.2$ | $73.3 \pm 0.7$ | $78.7 \pm 0.8$ | 78.7 |
| MMD | $98.3 \pm 0.1$ | $65.6 \pm 0.7$ | $69.7 \pm 1.0$ | $75.7 \pm 0.9$ | 77.3 |
| DANN | $97.3 \pm 1.3$ | $63.7 \pm 1.3$ | $72.6 \pm 1.4$ | $74.2 \pm 1.7$ | 76.9 |
| CDANN | $97.6 \pm 0.6$ | $63.4 \pm 0.8$ | $70.5 \pm 1.4$ | $78.6 \pm 0.5$ | 77.5 |
| MTL | $97.6 \pm 0.6$ | $60.6 \pm 1.3$ | $71.0 \pm 1.2$ | $77.2 \pm 0.7$ | 76.6 |
| SagNet | $97.3 \pm 0.4$ | $61.6 \pm 0.8$ | $73.4 \pm 1.9$ | $77.6 \pm 0.4$ | 77.5 |
| ARM | $97.2 \pm 0.5$ | $62.7 \pm 1.5$ | $70.6 \pm 0.6$ | $75.8 \pm 0.9$ | 76.6 |
| VREx | $96.9 \pm 0.3$ | $64.8 \pm 2.0$ | $69.7 \pm 1.8$ | $75.5 \pm 1.7$ | 76.7 |
| RSC | $97.5 \pm 0.6$ | $63.1 \pm 1.2$ | $73.0 \pm 1.3$ | $76.2 \pm 0.5$ | 77.5 |

### C.2.4 PACS

| Algorithm | A | C | P | S | Avg |
|---|---|---|---|---|---|
| ERM | $83.2 \pm 1.3$ | $76.8 \pm 1.7$ | $97.2 \pm 0.3$ | $74.8 \pm 1.3$ | 83.0 |
| IRM | $81.7 \pm 2.4$ | $77.0 \pm 1.3$ | $96.3 \pm 0.2$ | $71.1 \pm 2.2$ | 81.5 |
| GroupDRO | $84.4 \pm 0.7$ | $77.3 \pm 0.8$ | $96.8 \pm 0.8$ | $75.6 \pm 1.4$ | 83.5 |
| Mixup | $85.2 \pm 1.9$ | $77.0 \pm 1.7$ | $96.8 \pm 0.8$ | $73.9 \pm 1.6$ | 83.2 |
| MLDG | $81.4 \pm 3.6$ | $77.9 \pm 2.3$ | $96.2 \pm 0.3$ | $76.1 \pm 2.1$ | 82.9 |
| CORAL | $80.5 \pm 2.8$ | $74.5 \pm 0.4$ | $96.8 \pm 0.3$ | $78.6 \pm 1.4$ | 82.6 |
| MMD | $84.9 \pm 1.7$ | $75.1 \pm 2.0$ | $96.1 \pm 0.9$ | $76.5 \pm 1.5$ | 83.2 |
| DANN | $84.3 \pm 2.8$ | $72.4 \pm 2.8$ | $96.5 \pm 0.8$ | $70.8 \pm 1.3$ | 81.0 |
| CDANN | $78.3 \pm 2.8$ | $73.8 \pm 1.6$ | $96.4 \pm 0.5$ | $66.8 \pm 5.5$ | 78.8 |
| MTL | $85.6 \pm 1.5$ | $78.9 \pm 0.6$ | $97.1 \pm 0.3$ | $73.1 \pm 2.7$ | 83.7 |
| SagNet | $81.1 \pm 1.9$ | $75.4 \pm 1.3$ | $95.7 \pm 0.9$ | $77.2 \pm 0.6$ | 82.3 |
| ARM | $85.9 \pm 0.3$ | $73.3 \pm 1.9$ | $95.6 \pm 0.4$ | $72.1 \pm 2.4$ | 81.7 |
| VREx | $81.6 \pm 4.0$ | $74.1 \pm 0.3$ | $96.9 \pm 0.4$ | $72.8 \pm 2.1$ | 81.3 |
| RSC | $83.7 \pm 1.7$ | $82.9 \pm 1.1$ | $95.6 \pm 0.7$ | $68.1 \pm 1.5$ | 82.6 |

### C.2.5 OFFICEHOME

| Algorithm | A | C | P | R | Avg |
|---|---|---|---|---|---|
| ERM | 61.1 ± 0.9 | 50.7 ± 0.6 | 74.6 ± 0.3 | 76.4 ± 0.6 | 65.7 |
| IRM | 58.2 ± 1.2 | 51.6 ± 1.2 | 73.3 ± 2.2 | 74.1 ± 1.7 | 64.3 |
| GroupDRO | 59.9 ± 0.4 | 51.0 ± 0.4 | 73.7 ± 0.3 | 76.0 ± 0.2 | 65.2 |
| Mixup | 61.4 ± 0.5 | 53.0 ± 0.3 | 75.8 ± 0.2 | 77.7 ± 0.3 | 67.0 |
| MLDG | 60.5 ± 1.4 | 51.9 ± 0.2 | 74.4 ± 0.6 | 77.6 ± 0.4 | 66.1 |
| CORAL | 64.5 ± 0.8 | 54.8 ± 0.2 | 76.6 ± 0.3 | 78.1 ± 0.2 | 68.5 |
| MMD | 60.8 ± 0.7 | 53.7 ± 0.5 | 50.2 ± 19.9 | 76.0 ± 0.7 | 60.2 |
| DANN | 60.2 ± 1.3 | 52.2 ± 0.9 | 71.3 ± 2.0 | 76.0 ± 0.6 | 64.9 |
| CDANN | 58.7 ± 2.9 | 49.0 ± 2.1 | 73.6 ± 1.0 | 76.0 ± 1.1 | 64.3 |
| MTL | 59.1 ± 0.3 | 52.1 ± 1.2 | 74.7 ± 0.4 | 77.0 ± 0.6 | 65.7 |
| SagNet | 63.0 ± 0.8 | 54.0 ± 0.3 | 76.6 ± 0.3 | 76.8 ± 0.4 | 67.6 |
| ARM | 58.7 ± 0.8 | 49.8 ± 1.1 | 73.1 ± 0.5 | 75.9 ± 0.1 | 64.4 |
| VREx | 57.6 ± 3.4 | 51.3 ± 1.3 | 74.9 ± 0.2 | 75.8 ± 0.7 | 64.9 |
| RSC | 61.6 ± 1.0 | 51.1 ± 0.8 | 74.8 ± 1.1 | 75.7 ± 0.9 | 65.8 |

### C.2.6 TERRAINCOGNITA

| Algorithm | L100 | L38 | L43 | L46 | Avg |
|---|---|---|---|---|---|
| ERM | 34.4 ± 5.6 | 38.1 ± 4.0 | 55.7 ± 1.0 | 37.4 ± 1.1 | 41.4 |
| IRM | 46.7 ± 1.8 | 40.9 ± 2.1 | 52.2 ± 3.3 | 24.9 ± 10.0 | 41.2 |
| GroupDRO | 45.2 ± 6.2 | 40.1 ± 2.0 | 55.8 ± 1.4 | 38.3 ± 4.2 | 44.9 |
| Mixup | 59.7 ± 1.5 | 41.3 ± 2.1 | 55.9 ± 0.8 | 37.9 ± 1.5 | 48.7 |
| MLDG | 51.0 ± 1.9 | 39.2 ± 0.2 | 56.2 ± 1.1 | 38.3 ± 2.4 | 46.2 |
| CORAL | 52.4 ± 7.2 | 39.7 ± 1.5 | 56.1 ± 0.9 | 37.1 ± 2.2 | 46.3 |
| MMD | 49.1 ± 2.2 | 42.0 ± 1.6 | 55.3 ± 1.9 | 39.5 ± 2.0 | 46.5 |
| DANN | 46.9 ± 3.9 | 38.8 ± 1.1 | 55.5 ± 1.4 | 36.2 ± 1.1 | 44.4 |
| CDANN | 43.9 ± 7.3 | 32.5 ± 4.4 | 41.0 ± 7.8 | 42.4 ± 1.8 | 39.9 |
| MTL | 42.8 ± 4.6 | 43.9 ± 1.1 | 55.5 ± 0.8 | 37.5 ± 1.9 | 44.9 |
| SagNet | 48.1 ± 2.4 | 47.1 ± 0.8 | 54.4 ± 1.1 | 39.1 ± 1.8 | 47.2 |
| ARM | 48.9 ± 5.3 | 34.4 ± 3.5 | 51.9 ± 0.8 | 35.4 ± 2.3 | 42.6 |
| VREx | 46.4 ± 1.4 | 25.5 ± 5.8 | 39.6 ± 12.8 | 37.8 ± 3.6 | 37.3 |
| RSC | 40.0 ± 1.3 | 32.1 ± 2.5 | 53.9 ± 0.5 | 34.2 ± 0.2 | 40.0 |

### C.2.7 DOMAINNET

| Algorithm | clip | info | paint | quick | real | sketch | Avg |
|---|---|---|---|---|---|---|---|
| ERM | 58.1 ± 0.3 | 17.8 ± 0.3 | 47.0 ± 0.3 | 12.2 ± 0.4 | 59.2 ± 0.7 | 49.5 ± 0.6 | 40.6 |
| IRM | 47.5 ± 2.7 | 15.0 ± 1.5 | 37.3 ± 5.1 | 10.9 ± 0.5 | 48.0 ± 5.4 | 42.3 ± 3.1 | 33.5 |
| GroupDRO | 47.2 ± 0.5 | 17.0 ± 0.6 | 33.8 ± 0.5 | 9.2 ± 0.4 | 51.6 ± 0.4 | 39.2 ± 1.2 | 33.0 |
| Mixup | 54.4 ± 0.6 | 18.0 ± 0.4 | 44.5 ± 0.5 | 11.5 ± 0.2 | 55.8 ± 1.1 | 46.9 ± 0.2 | 38.5 |
| MLDG | 58.3 ± 0.7 | 19.3 ± 0.2 | 45.8 ± 0.7 | 13.2 ± 0.3 | 59.4 ± 0.2 | 49.8 ± 0.3 | 41.0 |
| CORAL | 59.2 ± 0.1 | 19.5 ± 0.3 | 46.2 ± 0.1 | 13.4 ± 0.4 | 59.1 ± 0.5 | 49.5 ± 0.8 | 41.1 |
| MMD | 32.2 ± 13.3 | 11.0 ± 4.6 | 26.8 ± 11.3 | 8.7 ± 2.1 | 32.7 ± 13.8 | 28.9 ± 11.9 | 23.4 |
| DANN | 52.7 ± 0.1 | 18.0 ± 0.3 | 44.2 ± 0.7 | 11.8 ± 0.1 | 55.5 ± 0.4 | 46.8 ± 0.6 | 38.2 |
| CDANN | 53.1 ± 0.9 | 17.3 ± 0.1 | 43.7 ± 0.9 | 11.6 ± 0.6 | 56.2 ± 0.4 | 45.9 ± 0.5 | 38.0 |
| MTL | 57.3 ± 0.3 | 19.3 ± 0.2 | 45.7 ± 0.4 | 12.5 ± 0.1 | 59.3 ± 0.2 | 49.2 ± 0.1 | 40.6 |
| SagNet | 56.2 ± 0.3 | 18.9 ± 0.2 | 46.2 ± 0.5 | 12.6 ± 0.6 | 58.2 ± 0.6 | 49.1 ± 0.2 | 40.2 |
| ARM | 49.0 ± 0.7 | 15.8 ± 0.3 | 40.8 ± 1.1 | 9.4 ± 0.2 | 53.0 ± 0.4 | 43.4 ± 0.3 | 35.2 |
| VREx | 46.5 ± 4.1 | 15.6 ± 1.8 | 35.8 ± 4.6 | 10.9 ± 0.3 | 49.6 ± 4.9 | 42.0 ± 3.0 | 33.4 |
| RSC | 55.0 ± 1.2 | 18.3 ± 0.5 | 44.4 ± 0.6 | 12.2 ± 0.2 | 55.7 ± 0.7 | 47.8 ± 0.9 | 38.9 |

## C.2.8 AVERAGES

| Algorithm | ColoredMNIST | RotatedMNIST | VLCS | PACS | OfficeHome | TerraIncognita | DomainNet | Avg |
|---|---|---|---|---|---|---|---|---|
| ERM | 36.7 ± 0.1 | 97.7 ± 0.0 | 77.2 ± 0.4 | 83.0 ± 0.7 | 65.7 ± 0.5 | 41.4 ± 1.4 | 40.6 ± 0.2 | 63.2 |
| IRM | 40.3 ± 4.2 | 97.0 ± 0.2 | 76.3 ± 0.6 | 81.5 ± 0.8 | 64.3 ± 1.5 | 41.2 ± 3.6 | 33.5 ± 3.0 | 62.0 |
| GroupDRO | 36.8 ± 0.1 | 97.6 ± 0.1 | 77.9 ± 0.5 | 83.5 ± 0.2 | 65.2 ± 0.2 | 44.9 ± 1.4 | 33.0 ± 0.3 | 62.7 |
| Mixup | 33.4 ± 4.7 | 97.8 ± 0.0 | 77.7 ± 0.6 | 83.2 ± 0.4 | 67.0 ± 0.2 | 48.7 ± 0.4 | 38.5 ± 0.3 | 63.8 |
| MLDG | 36.7 ± 0.2 | 97.6 ± 0.0 | 77.2 ± 0.9 | 82.9 ± 1.7 | 66.1 ± 0.5 | 46.2 ± 0.9 | 41.0 ± 0.2 | 64.0 |
| CORAL | 39.7 ± 2.8 | 97.8 ± 0.1 | 78.7 ± 0.4 | 82.6 ± 0.5 | 68.5 ± 0.2 | 46.3 ± 1.7 | 41.1 ± 0.1 | 65.0 |
| MMD | 36.8 ± 0.1 | 97.8 ± 0.1 | 77.3 ± 0.5 | 83.2 ± 0.2 | 60.2 ± 5.2 | 46.5 ± 1.5 | 23.4 ± 9.5 | 60.7 |
| DANN | 40.7 ± 2.3 | 97.6 ± 0.2 | 76.9 ± 0.4 | 81.0 ± 1.1 | 64.9 ± 1.2 | 44.4 ± 1.1 | 38.2 ± 0.2 | 63.4 |
| CDANN | 39.1 ± 4.4 | 97.5 ± 0.2 | 77.5 ± 0.2 | 78.8 ± 2.2 | 64.3 ± 1.7 | 39.9 ± 3.2 | 38.0 ± 0.1 | 62.2 |
| MTL | 35.0 ± 1.7 | 97.8 ± 0.1 | 76.6 ± 0.5 | 83.7 ± 0.4 | 65.7 ± 0.5 | 44.9 ± 1.2 | 40.6 ± 0.1 | 63.5 |
| SagNet | 36.5 ± 0.1 | 94.0 ± 3.0 | 77.5 ± 0.3 | 82.3 ± 0.1 | 67.6 ± 0.3 | 47.2 ± 0.9 | 40.2 ± 0.2 | 63.6 |
| ARM | 36.8 ± 0.0 | 98.1 ± 0.1 | 76.6 ± 0.5 | 81.7 ± 0.2 | 64.4 ± 0.2 | 42.6 ± 2.7 | 35.2 ± 0.1 | 62.2 |
| VREx | 36.9 ± 0.3 | 93.6 ± 3.4 | 76.7 ± 1.0 | 81.3 ± 0.9 | 64.9 ± 1.3 | 37.3 ± 3.0 | 33.4 ± 3.1 | 60.6 |
| RSC | 36.5 ± 0.2 | 97.6 ± 0.1 | 77.5 ± 0.5 | 82.6 ± 0.7 | 65.8 ± 0.7 | 40.0 ± 0.8 | 38.9 ± 0.5 | 62.7 |

## C.3 MODEL SELECTION: TEST-DOMAIN VALIDATION SET (ORACLE)

### C.3.1 COLOREDMNIST

| Algorithm | +90% | +80% | -90% | Avg |
|---|---|---|---|---|
| ERM | 71.8 ± 0.4 | 72.9 ± 0.1 | 28.7 ± 0.5 | 57.8 |
| IRM | 72.0 ± 0.1 | 72.5 ± 0.3 | 58.5 ± 3.3 | 67.7 |
| GroupDRO | 73.5 ± 0.3 | 73.0 ± 0.3 | 36.8 ± 2.8 | 61.1 |
| Mixup | 72.5 ± 0.2 | 73.9 ± 0.4 | 28.6 ± 0.2 | 58.4 |
| MLDG | 71.9 ± 0.3 | 73.5 ± 0.2 | 29.1 ± 0.9 | 58.2 |
| CORAL | 71.1 ± 0.2 | 73.4 ± 0.2 | 31.1 ± 1.6 | 58.6 |
| MMD | 69.0 ± 2.3 | 70.4 ± 1.6 | 50.6 ± 0.2 | 63.3 |
| DANN | 72.4 ± 0.5 | 73.9 ± 0.5 | 24.9 ± 2.7 | 57.0 |
| CDANN | 71.8 ± 0.5 | 72.9 ± 0.1 | 33.8 ± 6.4 | 59.5 |
| MTL | 71.2 ± 0.2 | 73.5 ± 0.2 | 28.0 ± 0.6 | 57.6 |
| SagNet | 72.1 ± 0.3 | 73.2 ± 0.3 | 29.4 ± 0.5 | 58.2 |
| ARM | 84.9 ± 0.9 | 76.8 ± 0.6 | 27.9 ± 2.1 | 63.2 |
| VREx | 72.8 ± 0.3 | 73.0 ± 0.3 | 55.2 ± 4.0 | 67.0 |
| RSC | 72.0 ± 0.1 | 73.2 ± 0.1 | 30.2 ± 1.6 | 58.5 |

### C.3.2 ROTATEDMNIST

| Algorithm | 0 | 15 | 30 | 45 | 60 | 75 | Avg |
|---|---|---|---|---|---|---|---|
| ERM | 95.3 ± 0.2 | 98.7 ± 0.1 | 98.9 ± 0.1 | 98.7 ± 0.2 | 98.9 ± 0.0 | 96.2 ± 0.2 | 97.8 |
| IRM | 94.9 ± 0.6 | 98.7 ± 0.2 | 98.6 ± 0.1 | 98.6 ± 0.2 | 98.7 ± 0.1 | 95.2 ± 0.3 | 97.5 |
| GroupDRO | 95.9 ± 0.1 | 99.0 ± 0.1 | 98.9 ± 0.1 | 98.8 ± 0.1 | 98.6 ± 0.1 | 96.3 ± 0.4 | 97.9 |
| Mixup | 95.8 ± 0.3 | 98.7 ± 0.0 | 99.0 ± 0.1 | 98.8 ± 0.1 | 98.8 ± 0.1 | 96.6 ± 0.2 | 98.0 |
| MLDG | 95.7 ± 0.2 | 98.9 ± 0.1 | 98.8 ± 0.1 | 98.9 ± 0.1 | 98.6 ± 0.1 | 95.8 ± 0.4 | 97.8 |
| CORAL | 96.2 ± 0.2 | 98.8 ± 0.1 | 98.8 ± 0.1 | 98.8 ± 0.1 | 98.9 ± 0.1 | 96.4 ± 0.2 | 98.0 |
| MMD | 96.1 ± 0.2 | 98.9 ± 0.0 | 99.0 ± 0.0 | 98.8 ± 0.0 | 98.9 ± 0.0 | 96.4 ± 0.2 | 98.0 |
| DANN | 95.9 ± 0.1 | 98.9 ± 0.1 | 98.6 ± 0.2 | 98.7 ± 0.1 | 98.9 ± 0.0 | 96.3 ± 0.3 | 97.9 |
| CDANN | 95.9 ± 0.2 | 98.8 ± 0.0 | 98.7 ± 0.1 | 98.9 ± 0.1 | 98.8 ± 0.1 | 96.1 ± 0.3 | 97.9 |
| MTL | 96.1 ± 0.2 | 98.9 ± 0.0 | 99.0 ± 0.0 | 98.7 ± 0.1 | 99.0 ± 0.0 | 95.8 ± 0.3 | 97.9 |
| SagNet | 95.9 ± 0.1 | 99.0 ± 0.1 | 98.9 ± 0.1 | 98.6 ± 0.1 | 98.8 ± 0.1 | 96.3 ± 0.1 | 97.9 |
| ARM | 95.9 ± 0.4 | 99.0 ± 0.1 | 98.8 ± 0.1 | 98.9 ± 0.1 | 99.1 ± 0.1 | 96.7 ± 0.2 | 98.1 |
| VREx | 95.5 ± 0.2 | 99.0 ± 0.0 | 98.7 ± 0.2 | 98.8 ± 0.1 | 98.8 ± 0.0 | 96.4 ± 0.0 | 97.9 |
| RSC | 95.4 ± 0.1 | 98.6 ± 0.1 | 98.6 ± 0.1 | 98.9 ± 0.0 | 98.8 ± 0.1 | 95.4 ± 0.3 | 97.6 |

### C.3.3 VLCS

| Algorithm | C | L | S | V | Avg |
|---|---|---|---|---|---|
| ERM | 97.6 ± 0.3 | 67.9 ± 0.7 | 70.9 ± 0.2 | 74.0 ± 0.6 | 77.6 |
| IRM | 97.3 ± 0.2 | 66.7 ± 0.1 | 71.0 ± 2.3 | 72.8 ± 0.4 | 76.9 |
| GroupDRO | 97.7 ± 0.2 | 65.9 ± 0.2 | 72.8 ± 0.8 | 73.4 ± 1.3 | 77.4 |
| Mixup | 97.8 ± 0.4 | 67.2 ± 0.4 | 71.5 ± 0.2 | 75.7 ± 0.6 | 78.1 |
| MLDG | 97.1 ± 0.5 | 66.6 ± 0.5 | 71.5 ± 0.1 | 75.0 ± 0.9 | 77.5 |
| CORAL | 97.3 ± 0.2 | 67.5 ± 0.6 | 71.6 ± 0.6 | 74.5 ± 0.0 | 77.7 |
| MMD | 98.8 ± 0.0 | 66.4 ± 0.4 | 70.8 ± 0.5 | 75.6 ± 0.4 | 77.9 |
| DANN | 99.0 ± 0.2 | 66.3 ± 1.2 | 73.4 ± 1.4 | 80.1 ± 0.5 | 79.7 |
| CDANN | 98.2 ± 0.1 | 68.8 ± 0.5 | 74.3 ± 0.6 | 78.1 ± 0.5 | 79.9 |
| MTL | 97.9 ± 0.7 | 66.1 ± 0.7 | 72.0 ± 0.4 | 74.9 ± 1.1 | 77.7 |
| SagNet | 97.4 ± 0.3 | 66.4 ± 0.4 | 71.6 ± 0.1 | 75.0 ± 0.8 | 77.6 |
| ARM | 97.6 ± 0.6 | 66.5 ± 0.3 | 72.7 ± 0.6 | 74.4 ± 0.7 | 77.8 |
| VREx | 98.4 ± 0.2 | 66.4 ± 0.7 | 72.8 ± 0.1 | 75.0 ± 1.4 | 78.1 |
| RSC | 98.0 ± 0.4 | 67.2 ± 0.3 | 70.3 ± 1.3 | 75.6 ± 0.4 | 77.8 |

### C.3.4 PACS

| Algorithm | A | C | P | S | Avg |
|---|---|---|---|---|---|
| ERM | 86.5 ± 1.0 | 81.3 ± 0.6 | 96.2 ± 0.3 | 82.7 ± 1.1 | 86.7 |
| IRM | 84.2 ± 0.9 | 79.7 ± 1.5 | 95.9 ± 0.4 | 78.3 ± 2.1 | 84.5 |
| GroupDRO | 87.5 ± 0.5 | 82.9 ± 0.6 | 97.1 ± 0.3 | 81.1 ± 1.2 | 87.1 |
| Mixup | 87.5 ± 0.4 | 81.6 ± 0.7 | 97.4 ± 0.2 | 80.8 ± 0.9 | 86.8 |
| MLDG | 87.0 ± 1.2 | 82.5 ± 0.9 | 96.7 ± 0.3 | 81.2 ± 0.6 | 86.8 |
| CORAL | 86.6 ± 0.8 | 81.8 ± 0.9 | 97.1 ± 0.5 | 82.7 ± 0.6 | 87.1 |
| MMD | 88.1 ± 0.8 | 82.6 ± 0.7 | 97.1 ± 0.5 | 81.2 ± 1.2 | 87.2 |
| DANN | 87.0 ± 0.4 | 80.3 ± 0.6 | 96.8 ± 0.3 | 76.9 ± 1.1 | 85.2 |
| CDANN | 87.7 ± 0.6 | 80.7 ± 1.2 | 97.3 ± 0.4 | 77.6 ± 1.5 | 85.8 |
| MTL | 87.0 ± 0.2 | 82.7 ± 0.8 | 96.5 ± 0.7 | 80.5 ± 0.8 | 86.7 |
| SagNet | 87.4 ± 0.5 | 81.2 ± 1.2 | 96.3 ± 0.8 | 80.7 ± 1.1 | 86.4 |
| ARM | 85.0 ± 1.2 | 81.4 ± 0.2 | 95.9 ± 0.3 | 80.9 ± 0.5 | 85.8 |
| VREx | 87.8 ± 1.2 | 81.8 ± 0.7 | 97.4 ± 0.2 | 82.1 ± 0.7 | 87.2 |
| RSC | 86.0 ± 0.7 | 81.8 ± 0.9 | 96.8 ± 0.7 | 80.4 ± 0.5 | 86.2 |

### C.3.5 OFFICEHOME

| Algorithm | A | C | P | R | Avg |
|---|---|---|---|---|---|
| ERM | 61.7 ± 0.7 | 53.4 ± 0.3 | 74.1 ± 0.4 | 76.2 ± 0.6 | 66.4 |
| IRM | 56.4 ± 3.2 | 51.2 ± 2.3 | 71.7 ± 2.7 | 72.7 ± 2.7 | 63.0 |
| GroupDRO | 60.5 ± 1.6 | 53.1 ± 0.3 | 75.5 ± 0.3 | 75.9 ± 0.7 | 66.2 |
| Mixup | 63.5 ± 0.2 | 54.6 ± 0.4 | 76.0 ± 0.3 | 78.0 ± 0.7 | 68.0 |
| MLDG | 60.5 ± 0.7 | 54.2 ± 0.5 | 75.0 ± 0.2 | 76.7 ± 0.5 | 66.6 |
| CORAL | 64.8 ± 0.8 | 54.1 ± 0.9 | 76.5 ± 0.4 | 78.2 ± 0.4 | 68.4 |
| MMD | 60.4 ± 1.0 | 53.4 ± 0.5 | 74.9 ± 0.1 | 76.1 ± 0.7 | 66.2 |
| DANN | 60.6 ± 1.4 | 51.8 ± 0.7 | 73.4 ± 0.5 | 75.5 ± 0.9 | 65.3 |
| CDANN | 57.9 ± 0.2 | 52.1 ± 1.2 | 74.9 ± 0.7 | 76.2 ± 0.2 | 65.3 |
| MTL | 60.7 ± 0.8 | 53.5 ± 1.3 | 75.2 ± 0.6 | 76.6 ± 0.6 | 66.5 |
| SagNet | 62.7 ± 0.5 | 53.6 ± 0.5 | 76.0 ± 0.3 | 77.8 ± 0.1 | 67.5 |
| ARM | 58.8 ± 0.5 | 51.8 ± 0.7 | 74.0 ± 0.1 | 74.4 ± 0.2 | 64.8 |
| VREx | 59.6 ± 1.0 | 53.3 ± 0.3 | 73.2 ± 0.5 | 76.6 ± 0.4 | 65.7 |
| RSC | 61.7 ± 0.8 | 53.0 ± 0.9 | 74.8 ± 0.8 | 76.3 ± 0.5 | 66.5 |

### C.3.6 TERRAINCOGNITA

| Algorithm | L100 | L38 | L43 | L46 | Avg |
|-----------|------|-----|-----|-----|-----|
| ERM | $59.4 \pm 0.9$ | $49.3 \pm 0.6$ | $60.1 \pm 1.1$ | $43.2 \pm 0.5$ | 53.0 |
| IRM | $56.5 \pm 2.5$ | $49.8 \pm 1.5$ | $57.1 \pm 2.2$ | $38.6 \pm 1.0$ | 50.5 |
| GroupDRO | $60.4 \pm 1.5$ | $48.3 \pm 0.4$ | $58.6 \pm 0.8$ | $42.2 \pm 0.8$ | 52.4 |
| Mixup | $67.6 \pm 1.8$ | $51.0 \pm 1.3$ | $59.0 \pm 0.0$ | $40.0 \pm 1.1$ | 54.4 |
| MLDG | $59.2 \pm 0.1$ | $49.0 \pm 0.9$ | $58.4 \pm 0.9$ | $41.4 \pm 1.0$ | 52.0 |
| CORAL | $60.4 \pm 0.9$ | $47.2 \pm 0.5$ | $59.3 \pm 0.4$ | $44.4 \pm 0.4$ | 52.8 |
| MMD | $60.6 \pm 1.1$ | $45.9 \pm 0.3$ | $57.8 \pm 0.5$ | $43.8 \pm 1.2$ | 52.0 |
| DANN | $55.2 \pm 1.9$ | $47.0 \pm 0.7$ | $57.2 \pm 0.9$ | $42.9 \pm 0.9$ | 50.6 |
| CDANN | $56.3 \pm 2.0$ | $47.1 \pm 0.9$ | $57.2 \pm 1.1$ | $42.4 \pm 0.8$ | 50.8 |
| MTL | $58.4 \pm 2.1$ | $48.4 \pm 0.8$ | $58.9 \pm 0.6$ | $43.0 \pm 1.3$ | 52.2 |
| SagNet | $56.4 \pm 1.9$ | $50.5 \pm 2.3$ | $59.1 \pm 0.5$ | $44.1 \pm 0.6$ | 52.5 |
| ARM | $60.1 \pm 1.5$ | $48.3 \pm 1.6$ | $55.3 \pm 0.6$ | $40.9 \pm 1.1$ | 51.2 |
| VREx | $56.8 \pm 1.7$ | $46.5 \pm 0.5$ | $58.4 \pm 0.3$ | $43.8 \pm 0.3$ | 51.4 |
| RSC | $59.9 \pm 1.4$ | $46.7 \pm 0.4$ | $57.8 \pm 0.5$ | $44.3 \pm 0.6$ | 52.1 |

### C.3.7 DOMAINNET

| Algorithm | clip | info | paint | quick | real | sketch | Avg |
|-----------|------|------|-------|-------|------|--------|-----|
| ERM | $58.6 \pm 0.3$ | $19.2 \pm 0.2$ | $47.0 \pm 0.3$ | $13.2 \pm 0.2$ | $59.9 \pm 0.3$ | $49.8 \pm 0.4$ | 41.3 |
| IRM | $40.4 \pm 6.6$ | $12.1 \pm 2.7$ | $31.4 \pm 5.7$ | $9.8 \pm 1.2$ | $37.7 \pm 9.0$ | $36.7 \pm 5.3$ | 28.0 |
| GroupDRO | $47.2 \pm 0.5$ | $17.5 \pm 0.4$ | $34.2 \pm 0.3$ | $9.2 \pm 0.4$ | $51.9 \pm 0.5$ | $40.1 \pm 0.6$ | 33.4 |
| Mixup | $55.6 \pm 0.1$ | $18.7 \pm 0.4$ | $45.1 \pm 0.5$ | $12.8 \pm 0.3$ | $57.6 \pm 0.5$ | $48.2 \pm 0.4$ | 39.6 |
| MLDG | $59.3 \pm 0.1$ | $19.6 \pm 0.2$ | $46.8 \pm 0.2$ | $13.4 \pm 0.2$ | $60.1 \pm 0.4$ | $50.4 \pm 0.3$ | 41.6 |
| CORAL | $59.2 \pm 0.1$ | $19.9 \pm 0.2$ | $47.4 \pm 0.2$ | $14.0 \pm 0.4$ | $59.8 \pm 0.2$ | $50.4 \pm 0.4$ | 41.8 |
| MMD | $32.2 \pm 13.3$ | $11.2 \pm 4.5$ | $26.8 \pm 11.3$ | $8.8 \pm 2.2$ | $32.7 \pm 13.8$ | $29.0 \pm 11.8$ | 23.5 |
| DANN | $53.1 \pm 0.2$ | $18.3 \pm 0.1$ | $44.2 \pm 0.7$ | $11.9 \pm 0.1$ | $55.5 \pm 0.4$ | $46.8 \pm 0.6$ | 38.3 |
| CDANN | $54.6 \pm 0.4$ | $17.3 \pm 0.1$ | $44.2 \pm 0.7$ | $12.8 \pm 0.2$ | $56.2 \pm 0.4$ | $45.9 \pm 0.5$ | 38.5 |
| MTL | $58.0 \pm 0.4$ | $19.2 \pm 0.2$ | $46.2 \pm 0.1$ | $12.7 \pm 0.2$ | $59.9 \pm 0.1$ | $49.0 \pm 0.0$ | 40.8 |
| SagNet | $57.7 \pm 0.3$ | $19.1 \pm 0.1$ | $46.3 \pm 0.5$ | $13.5 \pm 0.4$ | $58.9 \pm 0.4$ | $49.5 \pm 0.2$ | 40.8 |
| ARM | $49.6 \pm 0.4$ | $16.5 \pm 0.3$ | $41.5 \pm 0.8$ | $10.8 \pm 0.1$ | $53.5 \pm 0.3$ | $43.9 \pm 0.4$ | 36.0 |
| VREx | $43.3 \pm 4.5$ | $14.1 \pm 1.8$ | $32.5 \pm 5.0$ | $9.8 \pm 1.1$ | $43.5 \pm 5.6$ | $37.7 \pm 4.5$ | 30.1 |
| RSC | $55.0 \pm 1.2$ | $18.3 \pm 0.5$ | $44.4 \pm 0.6$ | $12.5 \pm 0.1$ | $55.7 \pm 0.7$ | $47.8 \pm 0.9$ | 38.9 |

### C.3.8 AVERAGES

| Algorithm | ColoredMNIST | RotatedMNIST | VLCS | PACS | OfficeHome | TerraIncognita | DomainNet | Avg |
|-----------|--------------|--------------|------|------|------------|----------------|-----------|-----|
| ERM | $57.8 \pm 0.2$ | $97.8 \pm 0.1$ | $77.6 \pm 0.3$ | $86.7 \pm 0.3$ | $66.4 \pm 0.5$ | $53.0 \pm 0.3$ | $41.3 \pm 0.1$ | 68.7 |
| IRM | $67.7 \pm 1.2$ | $97.5 \pm 0.2$ | $76.9 \pm 0.6$ | $84.5 \pm 1.1$ | $63.0 \pm 2.7$ | $50.5 \pm 0.7$ | $28.0 \pm 5.1$ | 66.9 |
| GroupDRO | $61.1 \pm 0.9$ | $97.9 \pm 0.1$ | $77.4 \pm 0.5$ | $87.1 \pm 0.1$ | $66.2 \pm 0.6$ | $52.4 \pm 0.1$ | $33.4 \pm 0.3$ | 67.9 |
| Mixup | $58.4 \pm 0.2$ | $98.0 \pm 0.1$ | $78.1 \pm 0.3$ | $86.8 \pm 0.3$ | $68.0 \pm 0.2$ | $54.4 \pm 0.3$ | $39.6 \pm 0.1$ | 69.0 |
| MLDG | $58.2 \pm 0.4$ | $97.8 \pm 0.1$ | $77.5 \pm 0.1$ | $86.8 \pm 0.4$ | $66.6 \pm 0.3$ | $52.0 \pm 0.1$ | $41.6 \pm 0.1$ | 68.7 |
| CORAL | $58.6 \pm 0.5$ | $98.0 \pm 0.0$ | $77.7 \pm 0.2$ | $87.1 \pm 0.5$ | $68.4 \pm 0.2$ | $52.8 \pm 0.2$ | $41.8 \pm 0.1$ | 69.2 |
| MMD | $63.3 \pm 1.3$ | $98.0 \pm 0.1$ | $77.9 \pm 0.1$ | $87.2 \pm 0.1$ | $66.2 \pm 0.3$ | $52.0 \pm 0.4$ | $23.5 \pm 9.4$ | 66.9 |
| DANN | $57.0 \pm 1.0$ | $97.9 \pm 0.1$ | $79.7 \pm 0.5$ | $85.2 \pm 0.2$ | $65.3 \pm 0.8$ | $50.6 \pm 0.4$ | $38.3 \pm 0.1$ | 67.7 |
| CDANN | $59.5 \pm 2.0$ | $97.9 \pm 0.0$ | $79.9 \pm 0.2$ | $85.8 \pm 0.8$ | $65.3 \pm 0.5$ | $50.8 \pm 0.6$ | $38.5 \pm 0.2$ | 68.2 |
| MTL | $57.6 \pm 0.3$ | $97.9 \pm 0.1$ | $77.7 \pm 0.5$ | $86.7 \pm 0.2$ | $66.5 \pm 0.4$ | $52.2 \pm 0.4$ | $40.8 \pm 0.1$ | 68.5 |
| SagNet | $58.2 \pm 0.3$ | $97.9 \pm 0.0$ | $77.6 \pm 0.1$ | $86.4 \pm 0.4$ | $67.5 \pm 0.2$ | $52.5 \pm 0.4$ | $40.8 \pm 0.2$ | 68.7 |
| ARM | $63.2 \pm 0.7$ | $98.1 \pm 0.1$ | $77.8 \pm 0.3$ | $85.8 \pm 0.2$ | $64.8 \pm 0.4$ | $51.2 \pm 0.5$ | $36.0 \pm 0.2$ | 68.1 |
| VREx | $67.0 \pm 1.3$ | $97.9 \pm 0.1$ | $78.1 \pm 0.2$ | $87.2 \pm 0.6$ | $65.7 \pm 0.3$ | $51.4 \pm 0.5$ | $30.1 \pm 3.7$ | 68.2 |
| RSC | $58.5 \pm 0.5$ | $97.6 \pm 0.1$ | $77.8 \pm 0.6$ | $86.2 \pm 0.5$ | $66.5 \pm 0.6$ | $52.1 \pm 0.2$ | $38.9 \pm 0.6$ | 68.2 |

## C.4 RESULTS OF A LARGER PACS SWEEP WITH 100 HYPERPARAMETER TRIALS

| ERM, model selection: | A | C | P | S | Avg |
|---|---|---|---|---|---|
| training-domain | $86.6 \pm 0.8$ | $79.7 \pm 0.6$ | $96.6 \pm 0.4$ | $77.8 \pm 0.8$ | 85.2 |
| leave-one-out-domain | $86.4 \pm 1.1$ | $78.2 \pm 1.0$ | $96.8 \pm 0.2$ | $76.0 \pm 2.1$ | 84.4 |
| test-domain (oracle) | $89.3 \pm 0.3$ | $82.2 \pm 0.5$ | $97.6 \pm 0.2$ | $82.7 \pm 1.1$ | 88.0 |

