# OpenReview forum: "In Search of Lost Domain Generalization"
_ICLR.cc/2021/Conference — ICLR 2021 Poster_

### Official Review · AnonReviewer4 · 2020-10-27
**Extremely well-written, but often too implicit**

**Rating:** 7
**Confidence:** 4

**Review:**

### Summary

The paper investigates the success of domain generalization (DG) approaches developed in recent years. In doing so, the authors evaluate a large variety of the most successful recent variants under principled model selection schemes (training-domain and leave-out-out validation), and find that standard empirical risk minimization outperforms – or is at least comparable – to all recently proposed state-of-the-art competitors. In bringing together a large set of heterogeneous methods the work makes an interesting contribution to the current DG literature. At the same time, in its current form I found the manuscript to be lacking any sufficiently substantial recommendations on how to improve the currently available zoo of methods.

### Strengths

The manuscript contains a very rigorous and impressively detailed background section (moved to the appendix) as well as experimental evaluations of the most important recent methods in domain generalization. The methodological introduction is very well written and principled, and contrasts domain generalization against other learning setups (from generative learning all the way to continual learning, domain adaptation etc.).

The experimental section does a convincing job of comparing empirical risk minimization against the various DG competitors that have been developed in recent years (DIVA, RSC, DDAIG, etc.), and in particular makes important recommendations that should find their way into practical research around the theme of DG. The proposed DomainBed environment (code in supplementary materials) appears of high quality, although it is difficult to gauge whether its ease-of-use in terms of extending it to new approaches will convince future researcher to adopt it and incorporate it with their propositions.

All in all, I found the manuscript to be a compelling read that contains an interesting alternative viewpoint on the role and limitations of DG. That being said, I think more needs to be done to make this paper more inspiring and useful for the wider community, in particular with regards to its concreteness (see below).

### Weaknesses

As highlighted above the paper is extremely well-written. However, it often leans in directions that I find too implicit, and – in my opinion – unnecessarily so. For example: "selecting hyperparameters is a learning problem at least as hard as fitting the model (inasmuch as we may interpret any model parameter as a hyperparameter)" (page 3, final paragraph). While this is a stimulating sentence, as a reader I am left wondering how this is crucial to motivating the central elements of the paper. In addition, what is the average DG researcher going to extract from this? Left untouched, I would suspect such a statement might invite some additional criticism from the viewpoint of previous DG research: how can one possibly guarantee that the tuning as part of DomainBed is optimal for the large heterogeneous variety of DG approaches out there, and who is to say that this can't be improved upon?

While page 8 introduces a number of interesting and compelling open questions, I would encourage the authors to provide more guidance in this context, potentially in the form of some experimentation, or analysis of the large set of results (e.g. which methods have strengths where, etc.). Some (again, too implicit) comments are there – e.g. "Therefore, we believe there is great promise in DG algorithms able to self-evaluate and self-adapt during test time." – more concise propositions as to which ideas the DG community can consider would be extremely helpful here.

I think the final paragraph of the paper nicely summarizes what I find to be the central weakness of this manuscript: in asking what a good benchmark is, instead of proposing some sort of alternative or making a recommendation, the authors opt for an intellectually compelling quote by Proust. Sure it's lovely to read, but I would again caution that is going to be of limited practical usefulness.

In summary, I therefore remain somewhat unconvinced that the manuscript, at least in its current form that mainly resorts to criticism of other DG approaches (while undoubtedly warranted (!)), but otherwise steers clear of any concise recommendations for future improvements, is an optimal use of content for an 8 pages conference paper.

### Minor points

"Although challenging, DG is the best approximation to real prediction problems, where unforeseen distributional discrepancies between training and testing data are surely expected." (page 3, third paragraph). DG is an interesting problem, but I'd be a bit more cautious in elevating its role for ML predictions problems in the real-world. At least in its current form DG approximates real-world problems from a (constrained) direction, and the above statement doesn't yet synergize well will the paragraph that rightfully asks whether DG benchmarks "are [..] the right datasets?" (page 8). Some clarification around the restrictions of DG early on would be helpful here.

While an interesting analysis of the differences between learning scenarios and domain generalization has been demarcated clearly from other types of learning problems, a very brief mentioning of the difference between multi-task and multi-domain learning would be beneficial given there is some confusion around these terms in the current literature.

---

> ### Author Response · Authors · 2020-11-23
> **Response**
>
> Thank you very much for a detailed review. We'd like to address some of your comments:
>
> **“It is difficult to gauge whether its ease-of-use in terms of extending it to new approaches will convince future researcher to adopt it”**
>
> We are very excited to communicate that we have already seen engagement and use of DomainBed in several other manuscripts. Furthermore, there have been several researchers from the research community that have contributed algorithms, datasets, and bug fixes.
>
> **“this is a stimulating sentence, as a reader I am left wondering how this is crucial to motivating the central elements of the paper”**
>
>  We have removed this sentence in order to focus our discussion on the central point, which is that model selection in domain generalization is nontrivial because we do not know any information about the test distribution at training time.
>
> **“how can one possibly guarantee that the tuning as part of DomainBed is optimal for the large heterogeneous variety of DG approaches out there”**
>
> Although we’ve been very careful in our implementation, we agree that such a guarantee isn’t really possible! We’ve added a discussion to our paper explaining the limitations of our claims. In short, our negative result claims apply only to our specific implementation. Although fundamentally limited, we feel such a result is still meaningful because (1) our implementation outperforms previously published results, (2) we motivate and explain implementation details transparently in the paper, and (3) we are open-sourcing DomainBed, getting in touch with the original authors of the baselines for code reviews, and inviting the whole research community to improve our results. We feel this is the best way to achieve rigorous and reproducible results across a large variety of algorithms.
>
> **“more guidance” / “concise recommendations” / “limited practical usefulness”**
>
> Our single biggest practical recommendation to researchers is to use the DomainBed codebase, which provides a reproducible, realistic, and consistent experimental setting. We agree that the focus should be on concrete guidance, so we’ve removed the Proust quote and some of the more abstract discussions. The focus of this paper is establishing strong baseline results for others to compare to, arguing for specific methodology and implementation choices, and contributing high-quality algorithm implementations -- any philosophical discussions are secondary.
>
> **“I'd be a bit more cautious in elevating its role for ML predictions problems in the real-world”**
>
> We have toned down this sentence, clarifying that the attractiveness of the domain generalization approach lies in the fact that it allows distributional shifts at testing time, a shift that can happen in real world scenarios.
>
> **“Some clarification around the restrictions of DG early on would be helpful here.”**
>
> We have added a sentence warning about the limitations of the vanilla DG setup, arguing that in many real world scenarios there may be external variables informing about task relatedness (space, time, annotations) that the DG framework ignores.
>
> We thank you once again for your dedication and time in writing this careful review.

---

### Official Review · AnonReviewer1 · 2020-10-28
**Interesting paper**

**Rating:** 6
**Confidence:** 5

**Review:**

Summary: This paper critically re-examines research in domain generalisation (DG), ie building models that robustly generalise to out-of-distribution data. It observes that existing methods are hard to compare, in particular due to unclear hyper-parameter and model selection criteria. It introduces a common benchmark suite including a well designed model selection procedure, and re-evaluates existing methods on this suite. The results show that under such controlled evaluation, the benefit of existing DG methods over vanilla empirical risk minimisation (ERM) largely disappear. This raises the concern that existing DG methods might be over-tuned and hard to replicate. By releasing the controlled benchmark suite, future research progress can be more reliably measured.

Strength:
+ The paper highlights an important point. Comparing existing methods is indeed tricky and complicated by model selection and hyper-parameter selection issues, etc.  It makes good recommendations for practice, such as requiring that any DG method also specifies its model selection method.  (We knew this already, but it’s good to remind people explicitly!) Helpfully it specifies a few reasonable options for model selection criteria which future papers could refer to rather than inventing ad-hoc approaches.
+ A common benchmark with a pre-specified strategy for hyper-parameter/early-stopping could be very helpful for more reliably comparing and measuring research progress in future.
+ A significant amount of effort was expended running a large and properly comparable evaluation across all several existing methods.

Weakness:
1. Strength of claim & validation. This paper is in part making a very strong negative result claim that a wide range of existing methods don’t work when implemented “properly”. This might be true, but then there is onus on the paper to make sure that all of the evaluation details are completely watertight.
- Are 20 trials enough for random search on all these models? It would be good to show some evidence that performance has saturated at this point (e.g., that performance doesn’t improve further with an order of magnitude more search). It would be good to also show some specific hyper-parameters found by the random search, so experts in the specific algorithms shown can assess if something sensible was found. Do the discovered hyper-parameters correspond to ERM for those methods where ERM is a special case?
- Are the hyper-parameter choices and ranges (Tab 6) satisfactory? For example some algorithms I have expertise on include hyper-parameters not documented in Tab 6, and its not clear how these are set. As another example, the DG research in my group has used SGD with momentum, not Adam, due to better stability in our experience. It’s not clear how this change affects things.
- Re Tab 6. It’s not clear: When optimising the methods do you jointly optimize the Resnet hyper parameters (such as learning rate, batch size) with the DG-specific parameters (bottom half of table) for each individual DG method. Or do you optimize ResNet hyperparameters first for ERM and then fix those before optimising the DG hyper-parameters?
- Overall, while the benchmark should be a useful contribution anyway, to believe the specific numerical conclusions, we have to trust the authors did a really good job re-implementing everything. If this research project had been setup instead as a competition where DG developers submit algorithms according to the constraints imposed by the benchmark for independent evaluation, and still reached the sae conclusion, then it would be more believable.

2. Insight. If we accept the headline conclusion that ERM indeed outperforms all the existing methods, it would be really nice to have some insight into what went wrong along the way. Some of these were alluded to, but not properly analysed.  For example:
- Do the prior methods have visible benefit over ERM on the more commonly evaluated networks like ResNet-18, and that benefit just fails to transfer to ResNet-50? We don’t know, because the comparison is only made on ResNet-50.
- Do the prior methods have visible benefit over ERM in the absence of data augmentation, and not otherwise?
- How much of the negative result is due to “proper tuning” improving ERM compared to previous poorly tuned implementations of the baselines; versus worsening the previously over-tuned DG methods?
- What is the primary source of over-tuning in existing DG methods? Which dominates among model selection, hyper-parameter selection, dataset split?
- Insights such as these would make the paper more satisfying, as well as believability -- by identifying more precisely the factors behind the negative result. Without these, it feels a bit empty.

3. Minor:
- I don’t understand the final comment in “untestable assumptions” on Pg 8. “DG algorithms that self-evaluate and self-adapt at test time” seems wrong. Adapting at test-time per-se, is against the problem spec of DG and blurs into domain adaptation.

============ UPDATE =============

Thanks to the authors for their feedback. I appreciate the efforts on clarification and loose-end tying.

One outstanding thing to clarify to help us understand whether Claim 1 can be fully supported:
- AFAIK the Table 1 / Table 5 that underpin this claim are comparing numbers copied from previous papers with numbers generated from Domain Bed benchmark? However I suspect the splits are not the same. For example, some previous benchmarks have a fixed split by default, while I understood Domain Bed use multiple random splits? If so the numbers are not directly comparable, and it still may not be fair to make a strong claim that tuned ERM outperforms prior work.

---

> ### Author Response · Authors · 2020-11-23
> **Response**
>
> Thank you very much for your thorough review! We've made changes to our paper as a result and we'd like to address some of your points below. For each point, we've updated the paper to reflect our answer (or will do so in the camera-ready, once our final sweep completes).
>
> **“This paper is in part making a very strong negative result claim” / “Is this watertight?” / "we have to trust the authors did a really good job re-implementing everything"**
>
> Thank you for bringing this up! We feel that it’s a really important point.
>
> Our first main contribution (“Claim 1” in the paper) is a strong ERM baseline result which outperforms prior published results. The validity and usefulness of this result don’t depend on the quality of our implementation choices.
>
> Regarding “Claim 2” in the paper: As you suggest, it’s impossible to rigorously prove a broad negative claim, and we want to emphasize that we’re not attempting to do so. Our Claim 2 isn’t that existing methods don’t outperform ERM when implemented "properly” (which would be an ill-defined claim), but rather that they didn’t outperform ERM in our restricted experimental setting. Such restriction is a fundamental limitation of any negative claim, but we think our claim is nonetheless meaningful for three reasons: (1) our implementation can outperform previously published results, (2) we explain implementation details and motivate them carefully in the paper, and (3) we are open-sourcing DomainBed, getting in touch with the original authors of the baselines for code reviews, and inviting the whole research community to improve our results. We feel this is the best way to achieve rigorous and reproducible results across a large variety of algorithms.
>
> We’ve added a discussion to the paper which explains this and makes the limitations of our claims clear.
>
> **“Are 20 trials enough for random search on all these models?”**
>
> We have run experiments on a random subset of algorithms and datasets with 100 trials, showing no significant improvement. We plan to include these in a supplementary in the camera-ready version after the entire sweep completes.
>
> **“It would be good to also show some specific hyper-parameters found by the random search”**
>
> We now include a script in DomainBed to print best hyperparameters, and plan to include the output in the supplementary material of the camera-ready version upon completion of the entire sweep.
>
> **"Do the discovered hyper-parameters correspond to ERM for those methods where ERM is a special case?**
>
> Yes, in some cases, algorithms find their best performance with a hyper-parameter configuration that results in an ERM-like behaviour.
>
> **“Are the hyper-parameter choices and ranges (Tab 6) satisfactory?”  / “SGD with momentum”**
>
> We chose the ranges in Table 6 based on experience gained from doing many exploratory experiments for each algorithm. Could you give us examples of such missing hyper-parameters? We are eager to conduct more experiments and push the numbers of all algorithms as high as possible. We have tried SGD with momentum with no improvement.
>
> ** “When optimising the methods do you jointly optimize the Resnet hyper parameters?”**
>
> All hyper-parameters are optimized jointly. We make sure this is clear in the revised version.
>
> **“If this research project had been setup instead as a competition [...] it would be more believable.”**
>
> We think this is a great way forward! We’ve open-sourced DomainBed (which we actively maintain) and invited the entire domain generalization research community to build together the most rigorous, reproducible, and reliable set of experimental results. We’re excited that we’ve already received such contributions from multiple researchers. In a field full of discrepancies in experimental protocols and impossible-to-replicate results, we believe this is the way forward.
>
> **“Do the prior methods have visible benefit over ERM on the more commonly evaluated networks like ResNet-18”**
>
> Preliminary experiments show that all methods lose in performance when using a ResNet-18, but the ranking of methods does not vary. We will add this result to the camera-ready upon completion of the entire sweep.
>
> **“Do the prior methods have visible benefit over ERM in the absence of data augmentation, and not otherwise?”**
>
> Table 4 shows that data augmentation does not influence domain generalization performance greatly. We speculate that this is because the ResNet models are already pre-trained on ImageNet with data augmentations.
>
> **“How much of the negative result is due to “proper tuning” improving ERM [...] versus worsening the previously over-tuned DG methods?”**
>
> Table 5 in the Appendix suggests it is the former, since all the other numbers are copied verbatim from the corresponding research papers.

---

> > ### Author Response · Authors · 2020-11-23
> > **Response (cont.)**
> >
> > **"What is the primary source of over-tuning in existing DG methods?"**
> >
> > This is a question worthy of a series of interesting papers. We believe it could be a different mixture of factors for every particular case. Papers in domain generalization often ignore details about model selection, use only one seed (no error bars), and do not explore different splits of the available data. Evaluating models this way, together with an excessive “publish or perish” pressure, could translate in some cases in suboptimal experimental protocols. By attributing all of our experimental results to a specific commit in our repository, all of these pitfalls can be avoided, enabling exciting, rigorous, and collaborative research.
> >
> > **"'DG algorithms that self-evaluate and self-adapt at test time' seems wrong"**
> >
> > We agree that many realizations of these ideas require venturing outside the problem spec of DG, but we feel that this might be what’s required to achieve OOD generalization in realistic settings. However, we’ve removed this sentence from the draft since it’s highly speculative.

---

### Official Review · AnonReviewer2 · 2020-10-29
**A testbed good for domain generalization research**

**Rating:** 7
**Confidence:** 5

**Review:**

In this paper, the authors implement a test bed to evaluate domain generalization methods in a unified way. The works is important because current methods use different model selection approaches, which may not reflect the inherent properties of the DG algorithms.

Model selection is a fundamentally difficult problem in the presence of distribution shift. However, it was significantly ignored in previous works. It is nice to see that the authors provide three kinds of model selection methods. From the results, it seems that existing DG methods do not have a clear advantage over ERM even when test-domain validation test is used. Does this mean existing methods themselves are not good? Or the dataset might not be appropriate for DG? It seems hard even for human to generalize to new domains when given a small number of domains with many changing factors.

I have some questions regarding the test bed details.
1)	Did the authors implement the existing methods or use the source codes provided by the authors?
2)	The authors carefully implemented and tuned ERM, did the authors also tuned the other methods carefully? This may require a significant amount of work, because different methods may need different engineering tricks.

---

> ### Author Response · Authors · 2020-11-23
> **Response**
>
> Thank you so much for your review.
>
> Indeed, those are exciting questions. We believe it is necessary to establish new, meaningful benchmarks for domain generalization, especially some where ERM fails to generalize. We believe it is in those experimental conditions where we can better understand the advantages of domain generalization algorithms.
>
> Re. question 1: We reimplemented all methods from scratch, carefully and following original source code where available. Our goal is a clean, easy-to-extend codebase (available as supplementary material) with state-of-the-art performance.
>
> Re. question 2: We spent a roughly equal amount of effort implementing and tuning each method. Ultimately whether we tuned them “carefully enough” is subjective, but we note that (1) our implementations outperform previously published results, (2) we motivate and explain implementation details transparently in the paper, and (3) we are open-sourcing DomainBed, getting in touch with the original authors of the baselines for code reviews, and inviting the whole research community to improve our results.

---

### Official Review · AnonReviewer3 · 2020-10-29
**This paper delivers a very clear message "when carefully implemented and tuned, ERM outperforms many SoTA DG methods"**

**Rating:** 8
**Confidence:** 4

**Review:**

##########################################################################

Summary:


In this work, authors implement and tune 14 DG algorithms and compare them across 6 datasets with 3 model selection criteria, and they find (i) a careful implementation of ERM outperforms the SoTAs. (ii) no competitor can outperform ERM by more than one point (iii) model selection matters for DG.

##########################################################################

Reasons for score:


I believe "the well-tuned ERM outperforms many SoTA DG methods" may not surprise many researchers in this area, but I would very much like to see a paper delivering this message clearly. Thus, I recommend an acceptance.

##########################################################################

Pros:


1. For quite a while, ERM (when carefully implemented and tuned) being the SoTA in DG is "elephant in the room". At least, we should admit that, quite often, the "improvements" claimed by those DA methods are gone when switching from a weaker backbone (e.g., ResNet-18) to a stonger one (e.g., ResNet-50). A high-standard testing protocol is a very important contribution in DG research.

2. This work brings an open-sourced software for replicating the existing methods, and comparing the newly proposed ones in a consistent and realistic setting.

##########################################################################

Cons:


1. Domain-Net, as a much larger scale and more challenging dataset, could be considered.

2. Can you elaborate the last sentence of Claim 2, i.e.,  "our advice to DG practitioners is to use CORALwith a hyperparameter search distribution that allows ERM-like behavior".

##########################################################################

A typo: (Page 6) "Table 5.2" shows that using a ResNet-50 neural network architecture. I think it should be "Table 4"

---

> ### Author Response · Authors · 2020-11-23
> **Response**
>
> Thank you very much for your review! We've made a few changes in response to your comments:
>
> - The revised version now includes DomainNet; all of our conclusions still hold.
> - We’ve also updated the paper to clarify the meaning of our CORAL recommendation. CORAL achieved the highest average score in our experiments, but when its regularization coefficient is zero, CORAL falls back to ERM, which according to our analyses is a safe contender when approaching domain generalization problems.

---

### Decision · Program_Chairs · 2021-01-07
**Final Decision**

**Decision:**

Accept (Poster)

**Comment:**

This paper provides an interesting analysis on the research on Domain Generalization with main principles and limitations. The authors provide a strong rebuttal to address some comments pointed by reviewers. All the reviews are very positive.
Hence, I recommend acceptance.